# Atmospheric Oxidation in the Presence of Clouds during the Deep Convective Clouds and Chemistry (DC3) Study

William H. Brune[1], Xinrong Ren[2,3], Li Zhang[1], Jingqiu Mao[4], David O. Miller[1], Bruce E. Anderson[5], Donald R. Blake[6], Ronald C. Cohen[7], Glenn S. Diskin[5], Samuel R. Hall[8], Thomas F. Hanisco[9], L. Gregory Huey[10], Benjamin A. Nault[11,*], Jeff Peischl[12,13], Ilana Pollack[12,13,#], Thomas B. Ryerson[13], Taylor Shingler[14,15], Armin Sorooshian[16,17], Kirk Ullmann[8], Armin Wisthaler[18], and Paul J. Wooldridge[7]

[1]Department of Meteorology and Atmospheric Science, Pennsylvania State University, University Park, PA
[2]Department of Atmospheric and Oceanic Science, University of Maryland, College Park, MD
[3]Air Resources Laboratory, National Oceanic and Atmospheric Administration, College Park, MD
[4]Department of Chemistry and Biochemistry, University of Alaska, Fairbanks, Fairbanks, AK
[5]Chemistry and Dynamics Branch, NASA Langley Research Center, Hampton, VA
[6]Department of Chemistry, University of California, Irvine, CA
[7]Departments of Chemistry and Earth and Planetary Sciences, University of California, Berkeley, Berkeley, CA
[8]Atmospheric Chemistry Observations and Modeling Laboratory, National Center for Atmospheric Research, Boulder, CO
[9]Atmospheric Chemistry and Dynamics Branch, Goddard Space Flight Center, Greenbelt, MD
[10]School of Earth and Atmospheric Sciences, Georgia Institute of Technology, Atlanta, GA
[11]Department of Earth and Planetary Sciences, University of California, Berkeley, Berkeley, CA
[12]Cooperative Institute for Research in Environmental Sciences, University of Colorado, Boulder, CO
[13]Earth System Research Laboratory, National Oceanic and Atmospheric Administration, Boulder, CO
[14]Science Systems and Applications, Inc., Hampton, VA
[15]Atmospheric Composition Branch, NASA Langley Research Center, Hampton, VA
[16]Department of Chemical and Environmental Engineering, University of Arizona, Tucson, AZ
[17]Department of Hydrology and Atmospheric Sciences, University of Arizona, Tucson, AZ
[18]Department of Chemistry, University of Oslo, Oslo, Norway
* now at Cooperative Institute for Research in Environmental Sciences, University of Colorado, Boulder, CO
# now at Department of Atmospheric Science, Colorado State University, Ft. Collins, CO

*Correspondence to*: William H. Brune (whb2@psu.edu)

**Abstract.** Deep convective clouds are critically important to the distribution of atmospheric constituents throughout the troposphere but are difficult environments to study. The Deep Convective Clouds and Chemistry (DC3) study in 2012 provided the environment, platforms, and instrumentation to test oxidation chemistry around deep convective clouds and their impacts downwind. Measurements on the NASA DC-8 aircraft included those of the radicals hydroxyl (OH) and hydroperoxyl (HO$_2$), OH reactivity, and more than 100 other chemical species and atmospheric properties. OH, HO$_2$, and OH reactivity were compared to photochemical models, some with and some without simplified heterogeneous chemistry, to test the understanding of atmospheric oxidation as encoded in the model. In general, the agreement between the observed and modeled OH, HO$_2$, and OH reactivity were within the combined uncertainties for the model without heterogeneous chemistry and the model including heterogeneous chemistry with small OH and HO$_2$ uptake consistent with laboratory studies. This agreement is generally independent of the altitude, ozone photolysis rate, nitric oxide and ozone abundances, modeled OH reactivity, and

aerosol and ice surface area. For a sunrise to midday flight downwind of a nighttime mesoscale convective system, the observed ozone increase is consistent with the calculated ozone production rate. Even with some observed-to-modeled discrepancies, these results provide evidence that a current measurement-constrained photochemical model can simulate observed atmospheric oxidation processes to within combined uncertainties, even around convective clouds. For this DC3 study, reduction in the combined uncertainties would be needed to confidently unmask errors or omissions in the model chemical mechanism.

## 1 Introduction

Deep convective clouds alter the chemical composition of the middle and upper troposphere (Chatfield and Crutzen, 1984). At its base, a cloud ingests air containing volatile organic compounds (VOCs) and anthropogenic pollutants emitted into the atmospheric boundary layer, lifts it to the upper troposphere where it spreads into the anvil and is eventually mixed with the surrounding air, including some from the lower stratosphere. Inside the convective cloud, the chemical composition is transformed: boundary layer air is diluted with cleaner mid-latitude air; water soluble chemical species are scrubbed by contact with cloud particles; and nitrogen oxides are added by lightning. At the same time, shading in the cloud core extends the lifetime of certain photochemically active compounds. This transformed chemical composition profoundly alters the atmospheric oxidation in the upper troposphere.

Atmospheric oxidation is driven primarily by hydroxyl (OH), whose concentration is strongly dependent on ozone ($O_3$), especially in the upper troposphere (Logan et al., 1981). Ozone is a source of OH through its destruction by solar ultraviolet radiation, which produces an excited state oxygen atom that can react with water vapor to produce OH. Hydroxyl reactions with methane ($CH_4$) and VOCs produce oxygenated volatile organic compounds (OVOCs), including organic peroxyl ($RO_2$, where R = $CH_3$, $C_2H_5$, …) and hydroperoxyl ($HO_2$). At the same time, OH reacts with carbon monoxide to produce $HO_2$. The reaction of these peroxyls with NO creates new nitrogen dioxide ($NO_2$), which then absorbs solar radiation to produce NO and $O(^3P)$, and $O(^3P)$ combines with $O_2$ to form new $O_3$. In the absence of nitrogen oxides ($NO_x = NO+NO_2$), the formation of OH and $HO_2$ acts to destroy $O_3$ through OH production and the reactions $O_3+HO_2$ and $O_3+OH$. Thus, atmospheric oxidation is strongly dependent on the chemical composition of the air exiting deep convective clouds.

Deep convective clouds transform the chemical composition of the upper troposphere in several ways (Barth et al., 2015, and references therein). $NO_x$ produced by lightning can have mixing ratios of several parts per billion (ppbv) in the anvil downwind of convection (Ridley et al., 1996; Schumann & Huntrieser, 2007; Pollack et al., 2016; Nault et al., 2017). In addition to long-lived chemical species such $CH_4$, carbon monoxide (CO), and alkanes, short-lived chemical species such as isoprene and its reaction products or those found in fire plumes can be rapidly transported from the planetary boundary layer (PBL) by

convection into the upper troposphere (Apel et al., 2012; Apel et al., 2015). Convection also provides cloud particle liquid and solid surfaces, which can interact with gas-phase chemical species such as peroxides, potentially scrubbing some chemical species from the gas-phase, thereby altering the gas-phase chemistry and its products (Jacob, 2000; Barth et al., 2016). The mixture of organics and nitrogen oxides forms organic nitrates, which act as sinks for both organic and hydrogen radicals and nitrogen oxides (Nault et al., 2016). This mixture of organic and hydrogen peroxyl and nitrogen oxides is calculated to enhance upper tropospheric $O_3$ production in the range of a 2-15 ppbv day$^{-1}$ (Pickering et al., 1990; Ren et al., 2008; Apel et al., 2012; Olson et al., 2012). Much of upper tropospheric OH and $HO_2$ is produced by the photolysis of oxygenated chemical species such as formaldehyde ($CH_2O$) and peroxides, which are products of organic chemical species that were lofted into the upper troposphere (Jaeglé et al., 1997; Wennberg et al., 1998; Ren et al., 2008).

Aircraft observations of tropospheric OH and $HO_2$ have been compared to photochemical box models constrained by other simultaneous measurements (Stone et al., 2012). In the planetary boundary layer, measurements of $HO_2$ (Fuchs et al., 2011) and, in some instruments, measurements of OH (Mao et al., 2012) are affected by interferences due predominantly to high abundances of alkenes and aromatics. In the free troposphere where the abundances of these chemical species are much lower, measured and modeled OH and/or $HO_2$ often agreed to within their combined uncertainties, which are similar to those for ground-based studies (Chen et al., 2012; Christian et al., 2017), but in most of these studies, either modeled and measured OH or $HO_2$ inexplicably disagreed beyond combined model and measurement uncertainty for certain altitudes or chemical compositions (Mauldin et al., 1998; Faloona et al., 2000; Tan et al., 2001; Olson et al., 2004; Olson et al., 2006; Ren et al., 2008; Stone et al., 2010; Kubistin et al., 2010; Olson et al., 2012; Regelin et al., 2013).

Only a few studies included OH and $HO_2$ measurements to test the impact of deep convective clouds on atmospheric oxidation in the upper troposphere. During the First Aerosol Characterization Experiment (ACE-1), modeled OH was 40% greater than measured OH in clouds, possibly due to uptake of OH or $HO_2$ on the cloud particles (Mauldin et al., 1998). In a different study, downwind of persistent deep convective clouds over the United States, the ratio of measured-to-modeled $HO_2$ increased from approximately 1 below 8 km altitude to 3 at 11 km altitude, suggesting an unknown source of $HO_x$ (= $OH+HO_2$) coming from the nearby convection (Ren et al., 2008). During the African Monsoon Multidisciplinary Analyses (AMMA) campaign, daytime $HO_2$ observations were generally simulated with a photochemical steady-state model, but not in clouds, where modeled $HO_2$ greatly exceeded observed $HO_2$ (Commane et al.,2010), suggesting $HO_2$ uptake on liquid cloud drops.

Heterogeneous chemistry on aerosol can impact OH and $HO_2$ abundances (Burkholder et al., 2015 and references therein). Typical OH and $HO_2$ accommodation coefficients used in global models are 0.2 on aerosol particles and 0.4-1.0 on ice. Results from a study over the north Atlantic Ocean indicated that the lower observed-than-modeled $HO_2$ could be resolved by including heterogeneous $HO_2$ loss (Jaeglé et al., 2000), but this did not resolve the same difference in clear air. This result differs from an analysis of in-cloud measurements over the western Pacific, which provide evidence that uptake in ice clouds has little

impact on $HO_2$ (Olson et al., 2004). However in the same study, the observed-to-modeled $HO_2$ during liquid cloud penetrations was on average only 0.65, compared to 0.83 outside of clouds, and this difference depended upon both the duration of the cloud penetration and the liquid water content (Olson et al., 2006). The uptake of $HO_2$ in liquid cloud particles was also observed by Whalley et al. (2015).

Laboratory studies show that the $HO_2$ effective uptake coefficient on moist aerosol particles that contain copper is probably much less than 0.01 in the lower troposphere but may be greater than 0.1 in the upper troposphere (Thornton et al., 2008). However, other laboratory studies show that adding organics to the particles or lowering the relative humidity can reduce uptake coefficients (Lakey et al., 2015; Lakey et al., 2016). These values are generally lower than the $HO_2$ effective uptake coefficient assumed in global chemical transport models. On ice surfaces, the OH uptake coefficient is thought to be at least 0.1 and probably larger (Burkholder et al., 2016). In global models, metal catalyzed $HO_2$ destruction on aerosol particles can reduce global $HO_2$ in a way that is more consistent with observed OH and $HO_2$ (Mao et al., 2010). More recently, a global sensitivity analysis shows that modeled $HO_2$ is most sensitive to aerosol uptake at high latitudes (Christian et al. 2016), consistent with the conclusion of Mao et al. (2010).

In this paper, we focus on the comparison of measured and modeled OH and $HO_2$ near deep convective clouds and the implications of this comparison. Our goal is to test the understanding of atmospheric oxidation around deep convective clouds and in the cloud anvils. One possible effect is heterogeneous uptake of OH, $HO_2$, and $RO_2$ around and in these clouds, which could alter OH and $HO_2$. The data for this analysis were collected during the Deep Convective Clouds and Chemistry (DC3) study in 2012.

## 2 Measurement and modelling methods

### 2.1 The DC3 Study

Barth et al. (2015) provide a detailed description of the objectives, strategy, locations, instrument payloads, and modeling for DC3. DC3 was designed to quantify the link between the properties of deep convective clouds and changes in chemical composition in the troposphere. This section provides information on the aspects of DC3 that relate most directly to the measured and modeled OH, $HO_2$, and OH reactivity.

DC3 involved several heavily instrumented aircraft, ground-based dual Doppler radar, lightning mapping arrays, and satellites (Barth et al., 2015). Studies were generally focused on areas in Colorado, Texas/Oklahoma, and Alabama that had dual Doppler radar coverage to quantify cloud properties. The field study took place in May through June, 2012. The aircraft were based in Salina, Kansas, which was central to the three main target regions in Colorado, Texas/Oklahoma, and Alabama. The NASA DC-8 sampled deep convection six times in Colorado, four times in Texas/Oklahoma, and three times in Alabama. Typically

working with the NSF G-V aircraft, the DC-8 would sample the inflow region to a growing cumulus cloud and then after it formed, spiral up to the anvil height (~8-12 km) and join the NSF G-V in sampling the outflow region of the same convection.

In addition, during the night before 21 June, the outflow from a mesoscale convective system (MCS) in the Midwest spread over Iowa and Missouri and then into Illinois and Tennessee. The DC-8 sampled the outflow of this MCS starting at sunrise of 21 June, flying six legs, each approximately 400 km long, across the outflow roughly perpendicular to the wind and adjusting the downwind distance to account for the outflow velocity (Nault et al., 2016). The southern 2/3 of the first three legs was in a thin cirrus cloud and the rest was clear.

Most DC3 flights began late morning in order to be in position to sample near active deep convection occurring in late afternoon and concluded near dusk for safety. About 90% of the flight time occurred when the solar zenith angle was less than 85°. Only the photochemical evolution flight on 21 June began before dawn.

The NASA DC-8 aircraft was the only aircraft that had an instrument to measure OH and $HO_2$. The DC-8 payload was quite comprehensive, thus providing detailed chemical composition, particle characteristics, and meteorological parameters to constrain the photochemical box model that is used to compare observed and modeled OH and $HO_2$. Direct comparison between observed and modeled OH and $HO_2$ is valid because the lifetimes of OH and $HO_2$ are short, a few seconds or less for OH and a few 10's of seconds for $HO_2$. Thus, the analyses in this paper uses only measurements from the DC-8.

### 2.2 Measurement of Hydroxyl (OH) and Hydroperoxyl ($HO_2$)

OH and $HO_2$ were measured with the Penn State Airborne Tropospheric Hydrogen Oxides Sensor (ATHOS), which uses laser induced fluorescence (LIF) in low pressure detection cells (Hard et al., 1984). ATHOS is described by Faloona et al. (2004). Sampled air is pulled through a 1.5 mm pinhole into a tube that leads to two detection axes. The pressure varies from 12 hPa at low altitudes to 3 hPa aloft. The laser beam is passed 32 times through the detection region with a multipass cell set at right angles to the gated microchannel plate detector. As the air passes through a laser beam, OH absorbs the laser radiation (3 kHz repetition rate, 20 ns pulse length) and fluoresces. In the first 100 ns, the signal contains fluorescence as well as scattering from the walls, Rayleigh scattering, and clouds drops. OH fluorescence is detected from 150 ns to 700 ns after each laser pulse. OH is detected in the first axis; reagent nitric oxide (NO) is added before the second axis to convert $HO_2$ to OH, which is then detected by LIF. The laser wavelength is tuned on-resonance with an OH transition for 15 seconds and off-resonance for 5 seconds, resulting in a measurement time resolution of 20 seconds. The OH fluorescence signal is the difference between on-resonance and off-resonance signals. The ATHOS nacelle inlet is attached below a nadir plate of the forward cargo bay of the DC-8 and the lasers, electronics, and vacuum pumps were inside the forward cargo bay.

In clouds, cloud particles can be pulled into the detection system and remain intact enough to cause large, short scattering signals in the fluorescence channels randomly during on-resonance and off-resonance periods. Differencing these signals to find OH creates large positive and negative noise, which reduces the measurement precision by as much as a factor of five. When the background signal due to these cloud particles exceeded the average background signal by four standard deviations, the on-line and off-line data were removed from the data set before the analyses were performed. Less than 3% of the data were removed. The overall results for OH and $HO_2$ vary less than 4% for filtering between two and six standard deviations.

The instrument was calibrated on the ground both in the laboratory and during the field campaign. Different sizes of pinholes were used in the calibration to produce different detection cell pressures to mimic different altitudes. Monitoring laser power, Rayleigh scattering, and laser linewidth maintained this calibration in flight. For the calibration, OH and $HO_2$ were produced through water vapor photolysis by UV light at 184.9 nm. Absolute OH and $HO_2$ mixing ratios were calculated by knowing the 184.9 nm photon flux, which was determined with a Cs-I phototube referenced to a NIST-calibrated photomultiplier tube, the $H_2O$ absorption cross section, the $H_2O$ mixing ratio, and the exposure time of the $H_2O$ to the 184.9 nm light. The absolute uncertainty was estimated to be ±16% for both OH and $HO_2$ at a $1\sigma$ confidence level. The $1\sigma$ precision for a one-minute integration time during this campaign was about 0.01 parts per trillion by volume (pptv, equivalent to pmol mol$^{-1}$) for OH and 0.1 pptv for $HO_2$. Further details about the calibration process may be found in Faloona et al. (2004).

For environments with substantial amounts of alkenes and aromatics, ATHOS has interferences for both OH (Mao et al., 2012) and $HO_2$ (Fuchs et al., 2011). New ATHOS measurement strategies have minimized these interferences, but these strategies were not fully developed in time for DC3. However, recent missions have shown that the OH interference is significant only just above forests or cities and is negligible above the PBL. On the other hand, the deep convective clouds encountered in DC3 can lift short-lived VOCs that cause the $HO_2$ interference to the upper troposphere. Because the ATHOS was still sensitive to this $RO_2$ interference in DC3, we are not able to determine if this interference is affecting the $HO_2$ observations around and in these clouds. For OH, we will factor in the likelihood that the OH has an interference in the PBL above forests in the discussion comparing observed and modeled OH.

For $HO_2$, the correction method uses more than 1000 $RO_2$ chemical species modeled by MCMv3.3.1 and assumes that they are ingested into the detection flow tube without any wall loss. The model then calculates the resulting OH, which is what would be detected as $HO_2$. The calculated concentration of reactant NO is ~$3\times10^{13}$ cm$^{-3}$ and the reaction time was determined to be 3.7 ms, as verified by the $HO_2$ conversion rate measured in the laboratory. This calculation was repeated for each one-minute time step and this calculated interference was then subtracted from the observed $HO_2$, resulting in the $HO_2$ values reported here. Observed $HO_2$ was reduced by an average of 2%, with some peaks of 10%, both in the PBL and aloft. Because the model $RO_2$ mechanisms are uncertain, the uncertainty for this correction is estimated to be a factor of 2, which increases the absolute uncertainty for $HO_2$ from ±16% to ±20%, $1\sigma$ confidence.

## 2.3 Measurement of OH Reactivity

The OH reactivity is the sum of the product of OH reactants and their reaction rate coefficients with OH and is the inverse of the OH lifetime. It is directly measured by adding OH to the air flowing through a tube and then monitoring the decay of the logarithm of the OH signal as the reaction time between the OH addition and OH detection is increased (Kovacs and Brune, 2001). OH can also be lost to the tube walls, so the measured OH reactivity must be corrected for this wall loss. The OH reactivity can be determined from Eq. (1).

$$k_{OH} = -\frac{ln\left([OH]/[OH]_0\right)}{\Delta t} - k_{wall} \tag{1}$$

$[OH]_0$ is the initial OH concentration, $[OH]$ is the [OH] concentration after a reaction time $\Delta t$ between OH and its reactants, and $k_{wall}$ is the OH wall loss.

The OH reactivity is measured with the OH Reactivity instrument (OHR), which sits in a rack in the DC-8 forward cargo (Mao et al., 2009). Ambient air is forced into a flow tube (10 cm dia.) at a velocity of 0.3-0.7 m s$^{-1}$, flows past the pinhole of an OH detection system similar to the one used for ATHOS, and then is expelled out of the aircraft. In a movable wand in the center of the flow tube, OH is produced by the photolysis of water vapor by 185 nm radiation and then injected into the flow tube, mixing with the ambient air flow. As the wand is pulled back, the distance between the injected OH and the OH reactants in the air increases, resulting in the OH decay. The distance divided by the measured velocity gives the reaction time. The wand moves 10 cm in 12 s and then returns to the starting position, measuring a decay every 20 s.

The OHR calibration was checked in the laboratory before and after DC3 using several different known amounts of different chemical species. During a semi-formal OHR Intercomparison in Juelich Germany in October 2015, the OHR instrument was combined with a different laser, wand drive, and electronics, and despite the difficulties encountered, it was found to produce accurate OH reactivity measurements (Fuchs et al., 2017).

The uncertainty in the OH reactivity measurement consists of an absolute uncertainty and the uncertainty associated with the wall loss subtraction. Changes in the OHR instrument between ARCTAS and DC3 result in slightly different instrument operation, wall loss, and measurement uncertainties between this paper and Mao et al. (2009). The pressure dependence of the OH wall loss was measured in the laboratory using ultra zero air (99.999% pure) and was found to between 2 s$^{-1}$ and 4 s$^{-1}$ over the range pressures equivalent to 0 to 12 km altitude. The flow tube wall is untreated aluminium so that every OH collision with the wall results in complete OH loss no matter the environment or altitude. Mao et al. (2009) confirm the zero found in the laboratory agrees with the zero found in flight. From these laboratory calibrations, the estimated uncertainty in the wall

loss correction is $\pm 0.5$ s$^{-1}$ (1$\sigma$ confidence). When the OH reactivity is 1 s$^{-1}$, the combined uncertainty from the absolute uncertainty and the zero decay is $\pm 0.6$ s$^{-1}$, 1$\sigma$ confidence, which suggests that 90% of the measurements should be within $\pm 1.2$ s$^{-1}$ of the mean value.

Air entering the OHR flow tube was warmed by ~5$^\circ$C at altitudes below 2 km to by as much as 75$^\circ$C at 12 km. The flow tube pressure was 50 hPa greater than ambient due to the ram force pushing air through the flow tube. These temperature and pressure differences can affect the reaction rate coefficients for some OH reactants and thus change the OH reactivity. In order to compare observed and model-calculated OH reactivity, the model was run for both ambient conditions and for the OHR flow tube pressure and temperature, but the observed and model-calculated OH reactivity will be compared for the OHR flow
tube pressure and temperature.

**2.4 Measurement of other chemical species, photolysis frequencies, and other environmental variables**

Accurate measurements of other chemical species and environmental variables are critical for this comparison of measured and modeled OH, HO$_2$, and OH reactivity. The photolysis frequency measurements are particularly critical for DC3 because of all the time spent flying around clouds and in the deep convection anvil. A list of these measurements is given in Table 1
and is summarized in Barth et al. (2015; 2016) and Pollack et al. (2016). The list of measured chemical species includes CO, CH$_4$, N$_2$O, NO, NO$_2$, O$_3$, organic nitrates, alkanes, alkenes, aromatics, aldehydes, alcohols, and peroxides.

**2.5 Photochemical Box Model**

The photochemical box model used in this study is based on the Matlab-based modeling framework, the Framework for Zero-Dimensional Atmospheric Modeling (F0AM), which was developed and made freely available by Glenn Wolfe (Wolfe et al.,
2016). The gas-phase photochemical mechanism was Master Chemical Mechanism v3.3.1 (MCMv331) (Saunders et al., 2003; Jenkin et al., 2003). This model was constrained by all simultaneous measurements of chemical species, photolysis frequencies, and meteorological variables (Table 1) and then run to calculate OH, HO$_2$, and all reaction products that were not measured, such as organic peroxy radicals. Pernitric acid (HO$_2$NO$_2$) was measured but was not used to constrain the model because few measurements were reported below 4 km. Measured and modeled HO$_2$NO$_2$ agree to within ~25% from 5 to 9 km, and modeled
HO$_2$NO$_2$ is 1.7 times that measured above 8 km. This difference between using modeled and observed HO$_2$NO$_2$ made only a few percent difference in modeled OH and HO$_2$.

A publically available merge file provided the constraining measurements for the photochemical model (DC3 Data, 2017). We chose the one-minute merge as a compromise between higher frequency measurements that needed to be averaged into one
minute bins and lower frequency measurements that needed to be interpolated between one minute bins. OH and HO$_2$ measurements, made every 20 seconds, were averaged into the one minute bins.

Heterogeneous chemistry was added to the model for some model runs. While many chemical species undergo heterogeneous chemistry, most of the chemical species that strongly influence OH and $HO_2$ were measured so that their heterogeneous chemistry can be ignored in these comparisons between observed and modeled OH and $HO_2$. However, organic peroxyl radicals were not measured and their heterogeneous chemistry could have an influence on OH and $HO_2$. Thus, heterogeneous chemistry is implemented in the model for OH, $HO_2$, and $RO_2$, even though the uptake of $RO_2$ is thought to be small.

The common types of particles encountered with convection are humidified submicron aerosol particles around and in the convection, liquid drops in lower altitude clouds, and ice particles in the deep convective cloud anvil. The effective uptake of OH, $HO_2$, and $RO_2$ onto these surfaces was found from Eq. (2).

$$\frac{1}{\gamma_{eff}} = \left(\gamma_{surface} + \left(\frac{1}{\alpha} + \frac{1}{\gamma_{sol} + \gamma_{rxn}}\right)^{-1}\right)^{-1} + \frac{0.75 + 0.286\,Kn}{Kn\,(Kn+1)} \tag{2}$$

$\gamma_{eff}$ is the effective uptake coefficient, $\gamma_{surface}$ is surface reaction uptake, $\alpha$ is the accommodation coefficient, $\gamma_{sol}$ is uptake due to diffusion through the liquid, $\gamma_{rxn}$ is uptake due to aqueous-phase reactions, and the last term is the inverse of the gas-phase diffusion in terms of the Knudsen number, $Kn$ (Burkholder et al., 2016; Tang et al., 2014). The first term on the right-hand side is the total uptake coefficient, $\gamma_{total}$. The gas-phase molecular diffusion of chemical species to the particle surface can limit the uptake, especially for large accommodation coefficients and large particles.

The total uptake coefficient for OH and $HO_2$ depends on the particle chemical composition, phase, and size (Burkholder et al., 2015). While some particle properties were measured in DC3, there are unknowns in particle composition and uncertainties in trying to calculate the uptake coefficient for each one-minute time step. The goal of including heterogeneous chemistry in the model is to determine if heterogeneous chemistry has a substantial impact on the comparison between the observed and modeled OH and $HO_2$. Thus, we will run the model with fixed values for the total uptake; if the impact on the modeled OH and $HO_2$ is substantial, then we will have to improve the parameterization of the total uptake.

For aerosol particles, the dry aerosol particle radius is multiplied by the growth factor and then the area-weighted median ambient aerosol radius is determined. The surface area is found by summing the surface area in each bin, multiplied by the bin width, and then corrected by the square of the growth factor for each minute. Ice particles are assumed to be spherical and their size distribution is used to determine the median particle radius. The surface area per $cm^{-3}$ of air is determined by summing the surface area per $cm^{-3}$ of air times the bin width, which were provided in the merge file for each minute.

The model is then run for three primary different cases: gas-phase with no heterogeneous chemistry (called "no-het"), heterogeneous chemistry (called "het") with $\alpha_{aer} = 0.2$ (which is consistent with the value used in some global models) and $\alpha_{ice}$

= 1.0 for OH, HO$_2$, and RO$_2$, and maximum heterogeneous chemistry (called "hetmax") with $\alpha_{aer}$ = 1.0 and $\alpha_{ice}$ = 1.0. The observed and modeled OH and HO$_2$ are compared for these three cases.

OH reactivity was also modeled for comparison to observed OH reactivity. Modeled OH reactivity was calculated from the measured chemical species plus OH reactants that were not measured but were produced by the photochemical model. Examples of these additional OH reactants are organic peroxy, organic peroxides, and unmeasured aldehydes. Uncertainty in the modeled OH reactivity is estimated to be ±10%, 1$\sigma$ confidence (Kovacs et al., 2001)

Uncertainty in the photochemical box model can be assessed with Monte Carlo methods in which model constraints are varied randomly over their uncertainty ranges and then the widths of the resulting distributions for OH and HO$_2$ abundances are used to determine the model uncertainties. If just reaction rate coefficients, photolysis frequencies, and reaction products are varied, the typical OH and HO$_2$ uncertainties at the 1$\sigma$ confidence level are typically ±(10-15)% (Thompson and Stewart, 1991; Kubiston et al., 2010; Olson et al., 2012; Regelin et al., 2013). When uncertainties in the measurements used to constrain the model are included, uncertainties at the 1$\sigma$ confidence level are typically ±20% or more for both local and global models (Chen et al., 2012; Christian et al., 2016). We use ±20% uncertainty at the 1$\sigma$ confidence level for the model uncertainty in this paper, which can be combined with the measurement uncertainty of about ±20% uncertainty at the 1$\sigma$ confidence level. We note that the observed-to-modeled difference for statistical significance lies approximately at the sum of the standard deviations of the mean for the observations and model. As a result, the factors of 1.4 and 1/1.4 serve as indicators for agreement between observed and modeled OH, HO$_2$, and OH reactivity.

We decided to make the comparisons between observations for all the DC3 results, including transits from Salina KS to the deep convection regions. Using all the data gives a more robust comparison and was found to give comparison results identical to that using data sets restricted to the radar-enhanced sites in Colorado, Texas/Oklahoma, and Alabama near vicinity of convection. The direct impact of deep convection is also tested by examining the observed-to-modeled comparison as a function of the ice surface area per cm$^{-3}$ of air. This analysis achieves the goals laid out in the introduction.

The model was run 27 times to test the sensitivity of the calculated OH and HO$_2$ to different factors. First, the chemical mechanism was expanded to include the reactions of CH$_3$O$_2$+OH and C$_2$H$_5$O$_2$+OH (Assaf et al., 2017), and in some cases, reactions of OH with the next 300 most significant RO$_2$ species that comprise 95% of the modeled RO$_2$ total, assuming a reaction rate coefficient of 10$^{-10}$ cm$^{-3}$s$^{-1}$. Adding the measured reactions decreased OH by ~1% (5% maximum) and increased HO$_2$ by <1% (15% maximum); adding the assumed reactions of OH with other RO$_2$ species changed modeled OH and HO$_2$ by less than 3%. Second, the decay frequency of the unconstrained modeled oxygenated intermediates was varied from 6 hours to 5 days. The resulting modeled OH and HO$_2$ varied by ~10% over this range. This decay time serves as a proxy for surface deposition in the planetary boundary layer as well as for recently measured rapid reactions of highly oxidized RO$_2$ + RO$_2$ to

form peroxides (Bernt et al., 2018). This decay time is highly uncertain. The mean of the model runs using decay times of 6 hours, 1 day, and 5 days and using the model mechanism including $CH_3O_2+OH$ and $C_2H_5O_2+OH$ was used for the comparisons to the measurements.

## 3 Results

The DC3 chemical environments dictate the OH and $HO_2$ abundances. Using all the flight data, altitude profiles of NO, CO, $O_3$, and particle surface area were collected into 0.5 km altitude bins and the median values were found (Fig. 1). The radar altitude is used for these comparisons because the surface elevations and thus planetary boundary layer (PBL) is at different pressure altitudes for the Colorado, Texas/Oklahoma, and Alabama regions.

Median NO exhibits a "C"-shaped profile with median values near 50 pptv in the planetary boundary layer, a minimum of 10 pptv at 5 km altitude, and a maximum value of 0.5 ppbv above 10 km. NO in the PBL was largely due to anthropogenic pollution; NO aloft was mainly due to a combination of lightning and stratospheric $NO_x$. Median CO slightly decreased from ~120 ppbv near the surface to ~80 ppbv at 12 km. Median $O_3$ was 50 ppbv in the PBL and then increased to as much as 100 ppbv near 12 km, likely due to stratospheric ozone influence. The surface area per $cm^{-3}$ of air of aerosol was $10^{-6}$ $cm^2cm^{-3}$ in

the PBL, but was about $3x10^{-7}$ $cm^2cm^{-3}$ above that. Median ice surface area was as much as $2x10^{-5}$ $cm^2cm^{-3}$, with high variability spanning a factor of 100.

## 3.1 Comparing $HO_x$ Observations with the no-het Model

Observations will be compared to the no-het model, which should be considered as the base case; the heterogeneous cases will be considered later. When observed and modelled OH and $HO_2$ are plotted against time, model results generally agree with

the observed values to within combined uncertainties, with occasional periods in which the observed values can be either much larger or smaller than all the model variants. A plot of this time series of observed and modeled OH and $HO_2$ includes fourteen model runs with model integration times varied from 3 to 24 hours, dilution frequencies varied from 6 hours to 5 days, and without or without including the $RO_2+OH$ reaction (Fig. S1).  Another way to examine these results is to plot the observed and modeled values as a function of altitude and other influential chemical species, photolysis frequency, or location.

Median observed OH was about $2x10^6$ $cm^{-3}$ from ~3 km to 10 km altitude, but was greater than $3x10^6$ $cm^{-3}$ below 3 km and just above 10 km, where a spike in OH brings the median OH to $3.5x10^6$ (Fig. 2). This profile is consistent with the NO profile; for the NO amounts measured in DC3, increased NO shifts more $HO_x$ to OH. The median OH observed-to-modeled ratio for the no-het case is close to 1.0 at 0-4 km, is 1.2-1.3 between 5 km and 9 km and 0.8-0.9 above 10 km. Generally, the observed-

to-modeled ratio is within the estimated combined uncertainties described previously. The statistics from scatter plots using a fitting routine that considered uncertainty in both the observations and the model (York et al., 2004) indicate generally good agreement between observed and modeled OH, with the model explaining 85% of the OH variance (Table 2, Fig. S2).

Median observed $HO_2$ was 20-22 pptv below 2 km altitude and decreased almost linearly to 3 pptv at 12 km (Fig. 3). This profile is consistent the altitude distribution of $HO_x$ sources, which are greater near the surface, and even with convective uplift of oxygenated chemical species such as HCHO, are still much less than at the surface. Again, the observed-to-modeled ratio is within the estimated combined uncertainties. In addition, the statistics from scatter plots indicate generally good agreement between observed and modeled $HO_2$, with the model explaining 68% of the $HO_2$ variance (Table 2, Fig. S2). However, the median observed-to-modeled ratio for the no-het case is 1.3 at 1 km altitude, decreases to ~1 at 5 km and remains near 1 above 5 km. The ratio of the modeled $RO_2$ to $HO_2$ is typically 1 in the PBL and 0.5 above 5 km. Error in our assumptions for the $RO_2$ interference in the $HO_2$ measurement is a possible cause of the greater observed-to-modeled $HO_2$ below 3 km, but this difference is still well within the combined uncertainty limits.

An indicator of the $HO_x$ cycling between OH and $HO_2$ is the $HO_2$/OH ratio. If the primary $HO_x$ production rate is smaller than the OH-$HO_2$ cycling rate and there is sufficient NO, the $HO_2$/OH ratio approximately equals the OH loss frequency that cycles OH into $HO_2$ divided by the reaction frequency of $HO_2$ reactions that cycle $HO_2$ to OH, primarily $HO_2$ reactions with NO and $O_3$. Both the observed and modelled $HO_2$/OH ratios are greater than 100 below 4 km and fall to less than 10 at 12 km (Fig. S3). This profile comes from the greater amount of OH reactants at lower altitudes, which increases the ratio, as opposed to the greater NO amount aloft, which decreases the $HO_2$/OH ratio. The observed-to-modeled ratio is within the approximate uncertainty limits (1σ confidence) of a factor of 1/1.4 to 1.4. .

Another good test of the model photochemistry is the comparison of observed and modeled OH and $HO_2$ as a function of controlling variables (Fig. 4). The photolysis frequency for $O_3$ producing an excited state O atom, $JO(^1D)$, and $O_3$ are both involved in the production of OH. $O_3$ and NO cycle $HO_2$ to OH, while modeled OH reactivity cycles OH back to $HO_2$. In general, measured and model OH and $HO_2$ agree from $2x10^{-6}$ $s^{-1}$ to $7x10^{-5}$ $s^{-1}$ for $JO(^1D)$, from $2x10^{-3}$ ppbv to $7x10^{-1}$ ppbv for NO, from 40 ppbv to 100 ppbv for $O_3$. For $JO(^1D)$ greater than $2x10^{-5}$ $s^{-1}$, the median observed-to-modeled $HO_2$ ratio is 0.98; the in-cloud ratio is an insignificant 10% less than in clear air, indicating that the observed photolysis frequency measurement is accurate even in clouds. The observed-to-modeled $HO_2$ ratio shows little evidence of a NO-dependence, although observed-to-modeled $HO_2$ exceeded 2 for ~2% of the values when NO was more than 0.5 ppbv. For the $O_3$ observations greater than 200 ppbv, which are 0.5% of all observations, the observed-to-modeled $HO_2$ and OH were both ~0.5. It is possible that the behavior as a function of controlling variables is also a function of altitude. However, with the exception of low values of $JO(^1D)$, the median observed-to-modeled OH and observed-to-modeled $HO_2$ are generally independent of both the controlling variables and altitude (Fig. S4). The observed-to-modeled OH and $HO_2$ are also independent of whether the measurements were made in Colorado, Texas/Oklahoma, or Alabama (Fig. S5), although the ratios for some altitudes vary widely due to fewer data points in the altitude medians.

## 3.2 Comparing OH Reactivity Observations with the no-het Model

The measured OH reactivity was typically 2 s$^{-1}$ to 5 s$^{-1}$ in the PBL but fell off to less than 1 s$^{-1}$ at 12 km altitude (Fig. 5). Observed and modeled OH reactivity are in reasonable agreement when considered in the context of the typical limit-of-detection for OH reactivity (Fuchs et al., 2017). For DC3, limit-of-detection for 20-second measurements is estimated to be about 0.6 s$^{-1}$, which means that most OH reactivity measurements were at or below the limit-of-detection. The modeled OH reactivity for the OHR flow tube temperature and pressure is greater than the modeled OH reactivity for ambient temperature and pressure by about 0.3 s$^{-1}$ below 2 km and by about 0.2 s$^{-1}$ above 10 km (Fig. 5). This difference is negligible below 2 km, but grows to a factor of 1.7 at 10 km where the ambient OH reactivity is just 0.2-0.3 s$^{-1}$. For the entire altitude, the difference between the observed and modeled OH reactivity is 0.2 s$^{-1}$ larger for the modelled OH reactivity at OHR flow tube conditions than for the modelled OH reactivity at ambient conditions, a difference that is swamped by the noise in the observed OH reactivity. The median observed OH reactivity is 1-3 s$^{-1}$ greater than the modeled OH reactivity below 3 km, and this difference could be evidence of missing OH reactivity.

The percent error between the OH reactivity calculated with modeled OH reactants and that calculated from only the measured OH reactants is, on average, less than 4%. In the PBL over Alabama, the OH reactivity calculated from the model after one day of integration was on average 10% larger than that calculated from the chemical species measurements. In the Colorado and Texas PBL, the average difference was less than 4%. According to the model calculations, CO contributes the most to the OH reactivity, with ~20% in the PBL and 30-40% aloft. Next is $CH_4$ at (5-10)%, HCHO at (5-10)%, $O_3$ at (2-10)%, $CH_3CHO$ at ~5%, and isoprene at (1-2)%, except in some PBL plumes where it was as much as 60%. The most significant ten chemical species were all measured and account for (60-70)% of the total model-calculated OH reactivity. Thus, for much of DC3, the OH reactivity calculated from the DC3 measurements almost completely comprises the measured OH reactivity.

## 3.3 Comparing OH Production and OH Loss

Another critical test of OH photochemistry is the balance between OH production and loss (Fig. 6). The OH lifetime is typically tenths of a second or less in the PBL and a few seconds at high altitude. Thus, for one-minute averages, OH production and loss should essentially be in balance to within the uncertainty estimates from a propagation of error analysis. Modeled OH production and loss are in balance. These uncertainty estimates were obtained by assuming that OH production is dominated by $HO_2+NO \rightarrow OH+NO_2$ and $O_3$ photolysis followed by reaction of excited state oxygen atoms with water vapor and that the OH loss is given by the OH reactivity multiplied by the OH concentration. The largest contributor to the uncertainty is the zero offset for the OH reactivity instrument. The observed OH loss is the observed OH multiplied by the observed OH reactivity, which was corrected for the difference between ambient and OHR flow tube conditions. While this analysis cannot preclude

missing or additional OH production and loss of $10^6$ cm$^{-3}$ s$^{-1}$ or less, the OH production and loss, which are calculated mostly from observations, agree to within uncertainties.

### 3.4 Comparing HO$_x$ Observations with Models Containing Heterogeneous Chemistry

Ample aerosol and ice during DC3 provide tests for possible effects of heterogeneous chemistry on OH and HO$_2$. We look first at the impact that adding heterogeneous chemistry (het, $\alpha_{ice} = 1.0$; $\alpha_{aer} = 0.2$), and maximum heterogeneous chemistry (hetmax, $\alpha_{ice} = 1.0$; $\alpha_{aer} = 1.0$) to the model has on the comparison of median observed and modeled OH and HO$_2$.

In Figures 2-4 and S6-S7 and Tables 2 and S1, the no-het and het models agree with the observed OH and HO$_2$ to within the combined uncertainties, with the exception of the het model for HO$_2$ when the aerosol surface area per cm$^{-3}$ of air was greater than $10^{-6}$ cm$^2$ cm$^{-3}$. In that case, the ratio of the observed-to-modeled HO$_2$ was too large, indicating that the het model was reducing HO$_2$ too much. On the other hand, the difference between the observations and hetmax model is greater than the uncertainty limits in almost every comparison. When the OH observed-to-modeled ratio is plotted as a function of aerosol surface area (Fig. 7), the no-het model gives a better agreement with the observations as a function of surface area than the het model does. For HO$_2$, the difference between the het model and the observations exceeds the combined uncertainty limits when the aerosol surface area per cm$^{-3}$ of air exceeds $10^{-6}$ cm$^2$cm$^{-3}$.

For heterogeneous uptake on ice, the observed-to-modeled comparisons of OH and HO$_2$ are roughly independent of ice surface area (Fig. S8). Comparing the model with HO$_2$ uptake on ice equalling 1 to a model with HO$_2$ uptake on ice set to zero decreases OH and HO$_2$ by only 10% for the largest ice surface area per cm$^{-3}$ of air of $7\times10^{-5}$ cm$^2$ cm$^{-3}$. HO$_2$ uptake on ice causes at most a few percent decrease in HO$_2$.

The addition of RO$_2$ heterogeneous chemistry on aerosol and ice did reduce the modeled RO$_2$ by 10% on average for the het model, but did not reduce either OH or HO$_2$ by more than 1%, on average. Including RO$_2$ uptake along with OH and HO$_2$ uptake had only a small impact on the heterogeneous effects on OH and HO$_2$.

### 3.5 Evolution of OH and HO$_2$ Downwind of a Nighttime Mesoscale Convective System (MCS)

The quasi-Lagrangian tracking of convective outflow on 21 June provided an opportunity to test the photochemical evolution of OH and HO$_2$ in the morning of 21 June. The flight legs perpendicular to the wind were approximately advected downwind. Initially, observed and modeled OH and HO$_2$ were close to 0 (Fig. 8). However, both modeled OH and HO$_2$ soon grew to exceed measured OH and HO$_2$ by about 20%-50% until about 8 am local time. For the remainder of the flight, observed and modeled OH is in substantial agreement for all three models, which are essentially identical except between 9:30 and 11:00 when the aerosol surface area per cm$^{-3}$ of air was the greatest. For HO$_2$, observed HO$_2$ exceeds modeled by a factor of ~1.5 after 9 (CST). In this time period, the observations agree best with the no-het and het models. Thus, the basic observed behavior

of OH and $HO_2$ is captured by the no-het and het models although there is quantitative disagreement between the model and observations of OH and $HO_2$ for local solar time between 5:45 and 8:15 (CST) and of $HO_2$ after 9 (CST).

Even though more than half of the first three legs were in an ice cloud, the scaled surface area for aerosol particles (Figure 8, top panel) and ice (Figure 8, bottom panel) do not correlate with the differences between observed and modeled OH and $HO_2$.

As a test of the photochemistry, OH can be calculated from the observed decay of OH reactants. Nault et al. (2016) showed that the observed decay of ethane, ethyne, and toluene were consistent with an average observed OH of $9.5x10^6$ $cm^{-3}$ for times from 7:35 to 9:50 local time. An ATHOS recalibration since the publication of Nault et al. (2016) brings the mean observed number for this time interval down to $8.8x10^6$ $cm^{-3}$. This recalibration was needed to account for the window absorption of calibrated 185 nm radiation that was neglected in the initial DC3 calibration. This median revised observed OH concentration is consistent with the mean modeled OH concentration of $9.6x10^6$ $cm^{-3}$. These results agree well within uncertainties and support the observed OH.

Another good test of the photochemistry is a comparison between the observed $O_3$ change and the accumulated calculated $O_3$ production rate (Fig. 9). The $O_3$ change is actually slightly larger than the observed $O_3$ because some of the new $O_3$ is partitioned into $NO_2$, so that the quantity of interest is the change in $O_3$ plus the change in $NO_2$. The ozone production rate, $P(O_3)$, can be calculated with Eq. (3).

$$P(O_3) = k_{NO+HO2}[NO][HO_2] + \sum_i k_{NO+RO2i}[NO][RO_{2i}] - L(O_3) \tag{3}$$

$k$ is a reaction rate coefficient and $L(O_3)$ is the loss terms for ozone. In this case, the loss term was only a few tenths ppbv $hr^{-1}$ (Fig. S9) and can be neglected. These rates are calculated in ppbv $min^{-1}$ and then accumulated at times from 5:45 to 11:15, in order to match the time period over which the DC-8 was sampling the MCS plume. This accumulated ozone production was calculated for the observed $HO_2$ plus modeled $RO_2$ and for the modeled $HO_2$ plus modeled $RO_2$. Modeled $RO_2$ is primarily $CH_3O_2$ and $CH_3CH_2O_2$ and its mixing ratio is half the $HO_2$ mixing ratio above 5 km. $HO_2$ accounts for a little more than half the total $O_3$ production. In order to compare the observed $O_3$ change to that accumulated from calculated $O_3$ production, an ozone offset of 53.4 ppbv was added to the accumulated $O_3$ production at 6:00 local solar time, which is when the ozone production commenced.

Ozone varied by 5-7 ppbv over the legs and was higher on one end of the leg than the other. These variations are smoothed using a 180-minute filter. During the 5 hours from 6:00 to 11:00, the observed $O_3$ change was 14 ppbv, although 2 ppbv of that change was in the final few minutes of observation. For the same time period, the ozone production calculated from modeled $HO_2$ and modeled $RO_2$ was 13 ppbv, and the ozone production calculated from observed $HO_2$ and modeled $RO_2$ was

13 ppbv. These three methods for determining $O_3$ production agree to well within their uncertainties and provide additional confirmation on the observed and modeled $HO_2$ and the modeled $RO_2$.

**4 Discussion**

These comparisons between median observed and modeled OH, $HO_2$, and OH reactivity display an agreement that is generally within the combined uncertainties of the observations and model. This agreement within uncertainties generally holds in scatter plots and associated statistics and as a function of altitude, $JO(^1D)$, NO, $O_3$, and the modeled OH reactivity. It also holds for the wide range of environments encountered during DC3, which includes different altitudes, cloudiness, and sunlight over roughly one third of the continental United States.

A closer look at the figures shows seemingly random differences between observed and modeled OH and $HO_2$. While most observed and modeled OH, $HO_2$, and OH reactivity one-minute data are within the combined uncertainties ($1\sigma$ confidence) of each other (57%), there are sometimes persistent unexplained differences (Fig. S1). An example is from the 21 June flight over Missouri and Illinois (Fig. 8). For both OH and $HO_2$, there are times when the modeled OH and $HO_2$ are higher than the measurements, lower than the measurements, and equal to the measurements. At the same time, the measured OH reactivity is sometimes greater than the modeled OH reactivity and sometimes equal to it. These types of discrepancies are proving to be hard to resolve. A simple correlation analysis found no strong correlation between the observed-to-modeled ratio or difference and any model output variable. Employing more sophisticated methods in future work may be more productive.

**4.1 Comparing DC3 to Previous Studies**

All previous aircraft studies that included measurements of OH and $HO_2$ have compared these observations to calculations from constrained photochemical box models. There is uncertainty in comparing the results from one study to another because often the instruments, their calibrations, and models are different or have evolved over time. Comparing results from very different environments amplifies this problem. As a result, the comparisons here are restricted to previous studies in air that was heavily influenced by convection.

NO, CO, $O_3$, and $JO(^1D)$ from DC3 are remarkably similar to those observed in INTEX-A in 2004 during flights over the central and eastern US (Ren et al., 2008) and to those observed from only 7 km to 10 km for HOOVER 2 in summer 2007 during flights over central Europe (Regelin et al., 2013). As in DC3, observed and modeled OH agree to within uncertainties for INTEX-A and HOOVER 2 over their altitude ranges. On the other hand, observed and modeled $HO_2$ agree to within uncertainties for DC3, HOOVER 2, and INTEX-A up to 8 km, but then observed $HO_2$ grows to exceed modeled $HO_2$ by a factor of 3 at 11 km for INTEX-A, in contrast with both DC3 and HOOVER 2 where observed and modeled $HO_2$ agree to within the combined uncertainties.

One explanation for the $HO_2$ discrepancy in INTEX-A could be the treatment of $HO_2NO_2$ formation rate in the model. MCMv3.3.1 uses a reaction rate coefficient that takes the much lower low-temperature reaction rate coefficient of the laboratory study of Bacak et al. (2011) into account, while the JPL evaluation (Burkholder et al., 2015) does not. For DC3, using the MCMv3.3.1 rate coefficient, observed-to-modeled OH and $HO_2$ are 0.85-0.9 at altitudes above 10 km altitude, and

5    modeled $HO_2NO_2$ is 1.9 times measured, but if the JPL recommended rate is used, then modeled OH and $HO_2$ both increase by a factor of 1.5 at altitudes above 10 km, shifting the observed-to-modeled ratios to 1.3-1.4, while increasing modeled $HO_2NO_2$ to 5-10 times that observed (Fig. S10). The JPL recommended reaction rate coefficient was used in the model for INTEX-A, which could explain some of the difference with DC3, except that observed-to-modeled OH ratio should also be a factor of 3 for INTEX-A.

For DC3, observed and modelled $HO_2$ appear to agree as a function of NO up to about 3 ppbv, which are the highest NO values encountered. For several previous ground-based studies, the observed $HO_2$ was not obviously greater than the modelled $HO_2$ until NO reached ~2 ppbv or greater (Martinez et al., 2003; Ren et al., 2003; Shirley et al., 2006; Kanaya et al., 2007; Brune et al., 2016). For aircraft studies, in some cases the observed $HO_2$ did not obviously exceed the modelled $HO_2$ until NO

approached 2 ppbv (Baier et al, 2017), while in other studies, the obvious exceedance occurred when NO was only a few hundred pptv (Faloona et al., 1999; Ren et al., 2008). Olson et al. (2006) showed that the Faloona et al. (1999) results for the SUCCESS campaign (central US, 1996) could be explained by the averaging of sharp plumes containing high NO and depleted $HO_2$ with the surrounding air. They showed that the SONEX (North Atlantic, 1997) results could be mostly explained by including all observed $HO_x$ precursors and updated kinetic rate coefficients and photolysis frequencies in the model. For

INTEX-A (Ren et al., 2008), the enhanced NO is in the upper troposphere, where the observed-to-modeled $HO_2$ reached a factor of 3. It is possible that the $HO_2$ calibration was in error at low pressure (i.e., higher altitudes), although observed and modelled $HO_2$ agree in the stratosphere. It is also possible that there were missing $HO_2$ sources or outdated reaction rates in the model chemistry. We intend to re-examine INTEX-A and other previous NASA DC-8 missions that included ATHOS to see if an updated model can better simulate these $HO_2$ observations.

The production rates of $HO_x$ (Fig. S11) and $O_3$ (Fig. S9) are comparable to those found in previous studies (Ren et al., 2008; Olson et al., 2012). Modeled $HO_x$ production is dominated by $O_3$ photolysis below 8 km, as has been previously observed (Regelin et al., 2013), but HCHO photolysis dominates above 10 km. $HO_x$ production by HONO photolysis essentially balances $HO_x$ loss by HONO formation. The calculated $O_3$ production rates are also comparable to these other studies, which find

production rates of a few ppbv $hr^{-1}$ in the PBL and above 10 km with a small loss at mid-altitude (Fig. S9). A spike in excess of 5 ppbv $hr^{-1}$ was found in a fire plume on 22 June (day-of-year 174), where abundances of both VOCs and $NO_x$ were enhanced.

## 4.2 Effects of Heterogeneous Chemistry on OH and HO₂

These DC3 results indicate that a total uptake coefficient is much less than 1 for aerosol. They also suggest that a total uptake coefficient of 0.2 is probably also too high because the model calculates $HO_2$ values that are much lower than observed $HO_2$ when the aerosol surface area per $cm^{-3}$ of air was greater than $10^{-6}$ $cm^2$ $cm^{-3}$. In the DC3 environment, $HO_x$ has substantial gas-phase loss pathways, making it difficult for heterogeneous chemistry to be dominant when the model uses an uptake coefficient consistent with laboratory studies, except at the largest aerosol surface area per $cm^{-3}$ of air. For the het case with a total uptake coefficient of 0.2, $HO_x$ loss including heterogeneous loss is 1.4 times the gas-phase only loss when the aerosol surface area per $cm^{-3}$ of air is $5x10^{-7}$ $cm^2 cm^{-3}$, but grows to 1.5 at $1x10^{-6}$ $cm^2 cm^{-3}$ and 2.5 for the few one-minute data points when it is greater than $3x10^{-6}$ $cm^2 cm^{-3}$. This increase in $HO_x$ loss translates into decrease in the modeled $HO_x$, with the decrease changing roughly as the square root of the increase in the $HO_x$ loss.

In DC3, the DC-8 spent hours flying in anvils of the cumulus clouds, which consisted of ice particles. DC3 provides evidence that the $HO_2$ uptake on ice is small. These results are consistent with $HO_2$ results over the western Pacific Ocean (Olson et al., 2004) but not with those over the northern Atlantic (Jeaglé et al., 2000). In Mauldin et al. (1998), a large difference between the observed and modeled OH was found in clouds, but this difference may have been due to the lack of photolysis frequency measurements, which are crucial to test photochemistry in a cloudy environment. In DC3, the DC-8 spent essentially no time in liquid clouds, for which there is evidence of measurable $HO_2$ uptake (Olson et al., 2006; Commane et al., 2010; Whalley et al., 2015). Thus these DC3 results provide constraints of $HO_2$ uptake on aerosol and ice particles, but not on liquid water particles.

In other cleaner environments or dirtier environments with much high aerosol surface area per $cm^{-3}$ of air, heterogeneous chemistry can be a substantial fraction of the entire $HO_x$ loss and thus affect $HO_2$ and OH abundances. Measuring OH and $HO_2$ in these environments will provide a better test of the understanding of $HO_2$ heterogeneous chemistry on aerosol particles than DC3 did.

## 5 Conclusions

The general agreement between the observed and modeled OH and $HO_2$ for the complex DC3 environment is encouraging. It suggests that a photochemical box model can simulate the observed OH and $HO_2$ to well within combined uncertainties, if properly constrained with measurements of other chemical species, photolysis frequencies, and environmental conditions. On the other hand, it is difficult to explain the unexpected deviations between observed and modeled OH and $HO_2$, such as is observed in Fig. 8 or Fig. S1. Neither heterogeneous chemistry nor organic peroxyl chemistry are able to explain these deviations.

There are other possible causes for these discrepancies. First, it can be difficult to maintain instrument calibrations for not only OH and $HO_2$ but also for all the other measurements that were used to constrain the model to calculate OH and $HO_2$. Second, the simultaneous measurements need to be properly conditioned so that they can be used as model constraints. This process includes filling in isolated missing values because, if this was not done, the constraining data set would be sparse. For DC3,

using the merged data set with no interpolation is less than 10% of the full data set, but the observed and modeled OH and $HO_2$ have essentially the same relationships as with the interpolated data set (Table S21). Third, the model parameters, such as integration times and decay times, must be set up so that the model calculations represent the observations and their variations. Varying these times caused a range of modeled values that was far smaller than the large observed-to-modeled differences, as seen in Fig. S1. For DC3, the one-minute data are adequate for the timescale of variations for most cases, except

in small fire plumes and some spikes in lightning $NO_x$ in the anvil. Fourth, multiple methods are needed to determine if differences between observations and model are significant. For DC3, the comparisons between observations and models are robust despite the method of comparison. Thus, none of these appear to be the cause of the unexplained deviations between observed and modeled OH and $HO_2$. A more thorough model uncertainty and sensitivity analysis could unveil the cause.

Even with these observed-to-modeled discrepancies, the general agreement for observed and modeled OH and $HO_2$ suggests that current photochemical models can simulate observed atmospheric oxidation processes even around clouds to within these combined uncertainties. Reducing these uncertainties will enable comparisons of observed and modeled OH and $HO_2$ to provide a more stringent test of the understanding of atmospheric oxidation chemistry and thus to lead to an improvement in that understanding.

**Code Availability**

The Matlab code used for the zero-dimensional photochemical box modelling with the MCMv3.3.1 mechanism can be downloaded from the website https://sites.google.com/site/wolfegm/models. The manuscript describing this model is Wolfe, G.M, Marvin, M.R., Roberts, S.J., Travis, K.R., and Liao, J.: The Framework for 0-D Atmospheric Modeling (F0AM) v3.1, Geosci. Model Dev., 9, 3309-3319, doi: 10.5194/gmd-9-3309-2016, 2016.

**Data Availability**

The merge file for the DC3 DC-8 data and the updated OH, $HO_2$, and OH reactivity numbers can be accessed from http://data. eol.ucar.edu/ or http://www-air.larc.nasa.gov/cgi-bin/ArcView/dc3-seac4rs, doi:10.5067/Aircraft/DC3/DC8/Aerosol-TraceGas

**Author Contribution**

All authors contributed DC3 measurements that are critical for the modeling and comparison of modeled and measured OH and $HO_2$ in this manuscript. W. Brune wrote the manuscript and made edits provided by the co-authors before submitting the manuscript.

5 **Competing Interests**

The authors declare that they have no conflicts of interest.

**Acknowledgments**

The Deep Convective Clouds and Chemistry (DC3) experiment is sponsored by the U.S. National Science Foundation (NSF), the National Aeronautics and Space Administration (NASA), the National Oceanic and Atmospheric Administration (NOAA), 10 and the Deutsches Zentrum fuer Luftund Raumfahrt (DLR). Archived field data can be accessed from http://data. eol.ucar.edu/ or http://www-air.larc. nasa.gov/cgi-bin/ArcView/dc3-seac4rs. Data provided by NCAR/EOL are supported by the National Science Foundation. Acetone/propanal measurements aboard the DC-8 during DC3 were supported by the Austrian Federal Ministry for Transport, Innovation, and Technology (BMVIT) through the Austrian Space Applications Programme (ASAP) of the Austrian Research Promotion Agency (FFG). Support for ATHOS and OHR measurements aboard the NASA DC-8 15 during DC3 comes from NASA grant NNX12AB84G. We thank NASA management, pilots, and operations personnel for the opportunity to gather these observations, the people of Salina, KS for hosting us and providing excellent facilities, G. Wolfe for his publically available F0AM model framework, and the University of Leeds for the publically available MCMv3.3.1 photochemical model. We also thank J. Thornton for his insights into $HO_2$ heterogeneous chemistry, and P. Lawson, P. Wennberg, J. Crounse, and J. St. Clair for the use of their measurements.

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

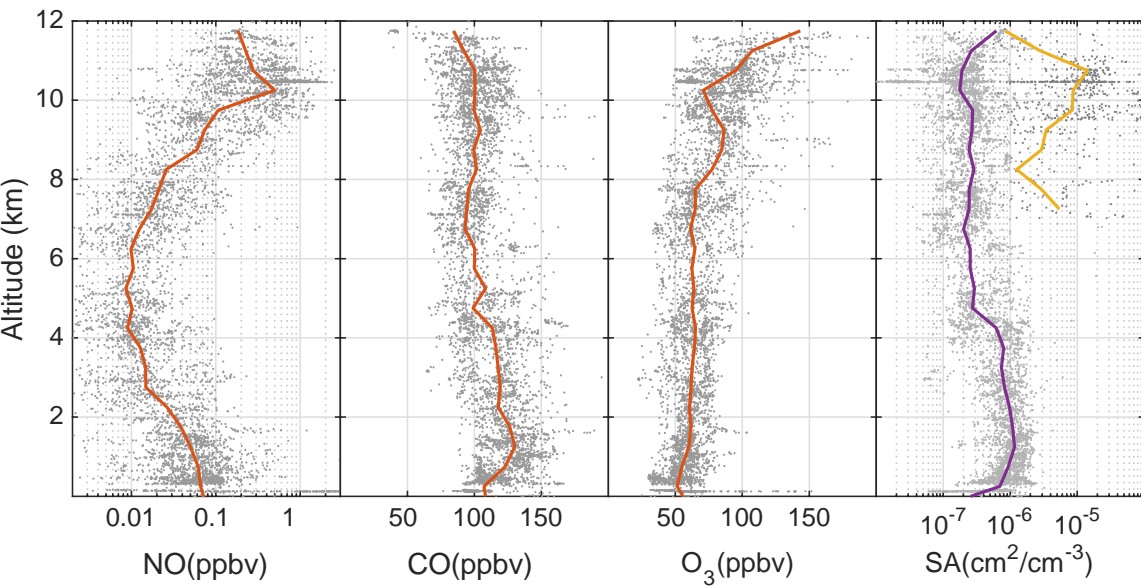

**Figure 1.** Altitude profiles of median NO, CO, O₃, and aerosol (purple) and ice (yellow) surface area (SA). Grey points are individual one-minute data, while median values are represented by the lines. These data are for the entire DC3 study.

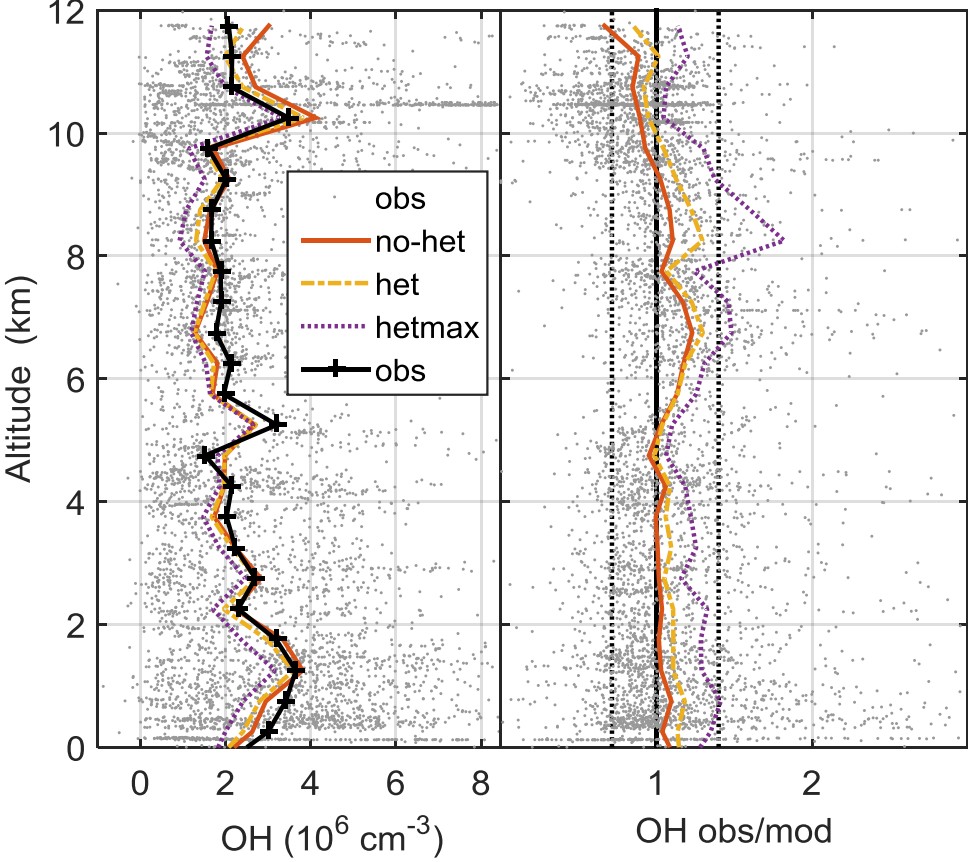

**Figure 2**. Median measured and modeled OH as a function of altitude (left); median ratio of observed-to-modeled OH as a function of altitude (right) for the three models. Gray points are individual 1-minute OH observations and ratios of observed-to-no-heterogeneous modeled OH. Dotted vertical black lines on right (1/1.4, 1.4) are approximate indicators of measurement-model agreement.

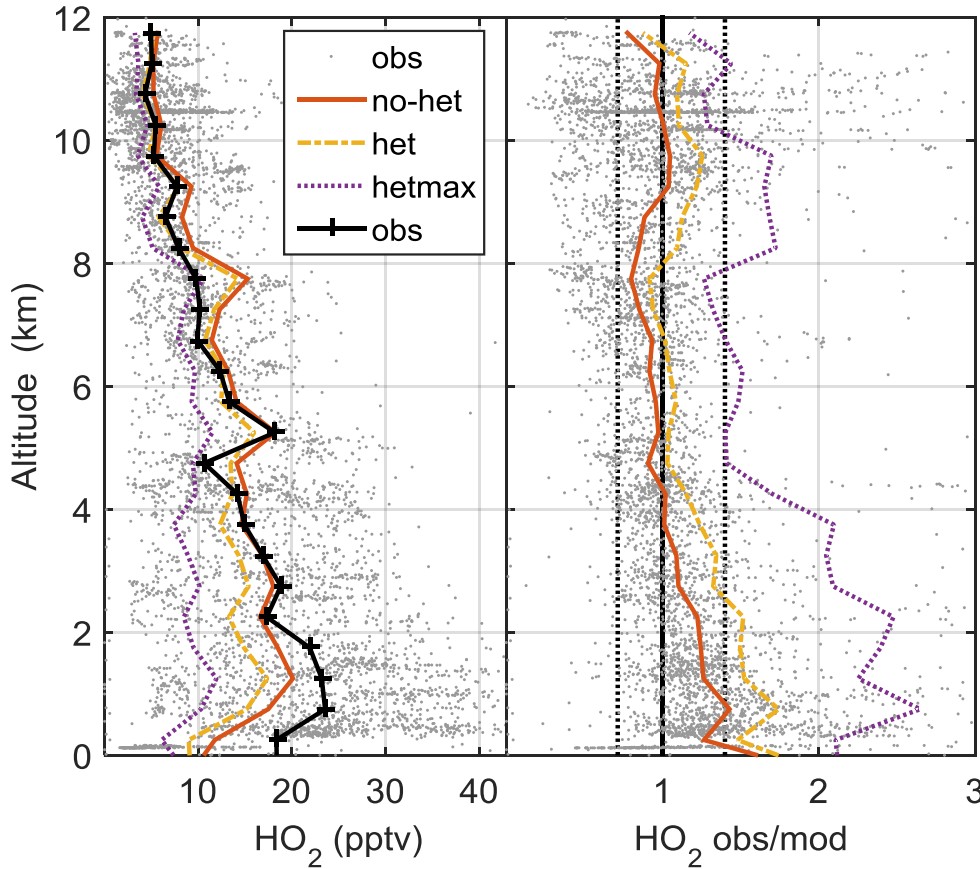

**Figure 3**. Median measured and modeled HO₂ as a function of altitude (left); median ratio of observed-to-modeled HO₂ as a function of altitude (right) for the three models. Gray points are individual 1-minute OH observations and ratios of observed-to-no-heterogeneous modeled OH. Dotted vertical black lines on right (1/1.4, 1.4) are approximate indicators of measurement-model agreement.

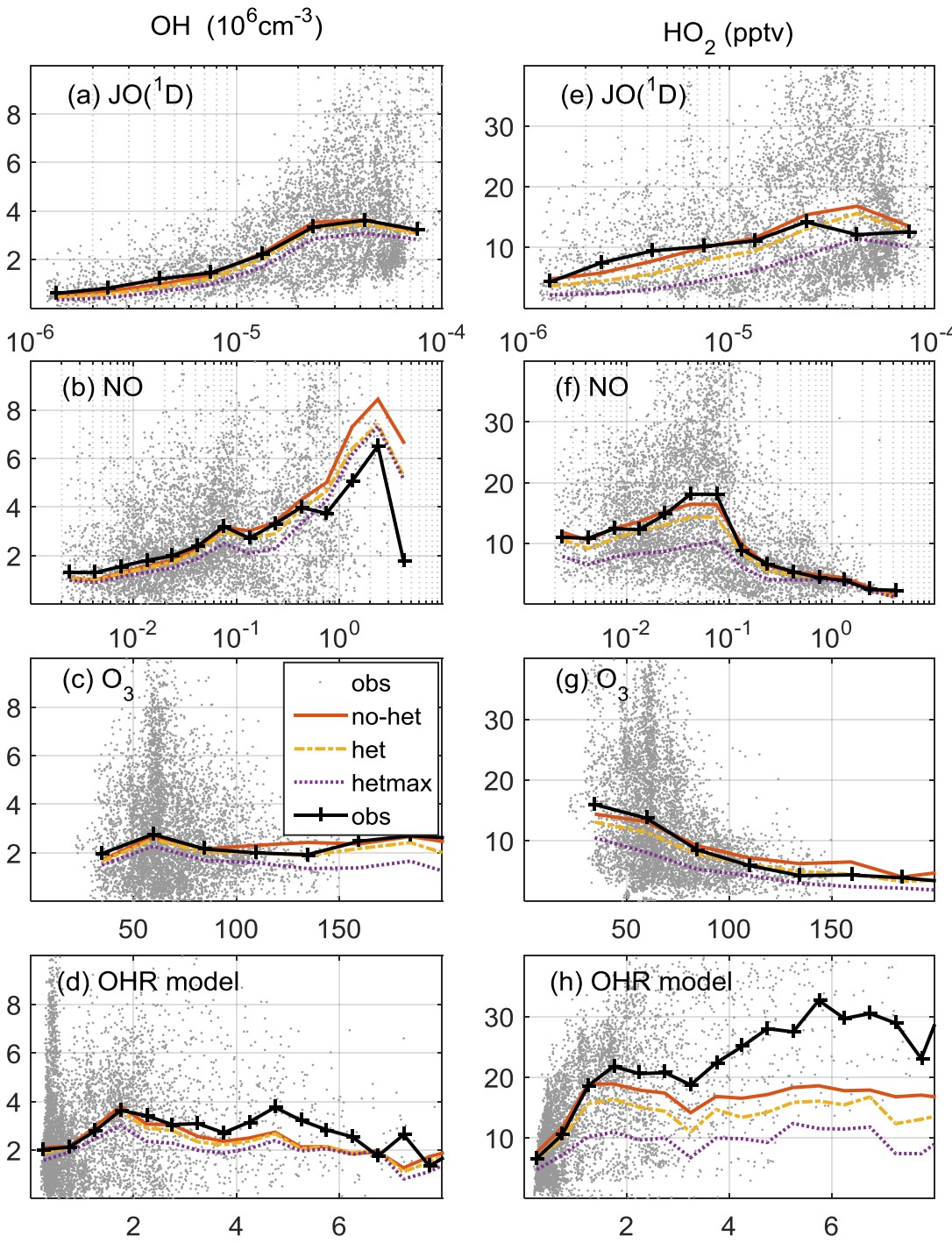

**Figure 4**. Measured and modeled OH (left) and HO$_2$ (right) as a function of controlling variables: JO($^1$D) in s$^{-1}$, NO in ppbv, O$_3$ in ppbv, and modeled OH reactivity in s$^{-1}$. On the left, OH is in units of $10^6$ cm$^{-3}$; on the right, HO$_2$ is in units of pptv.

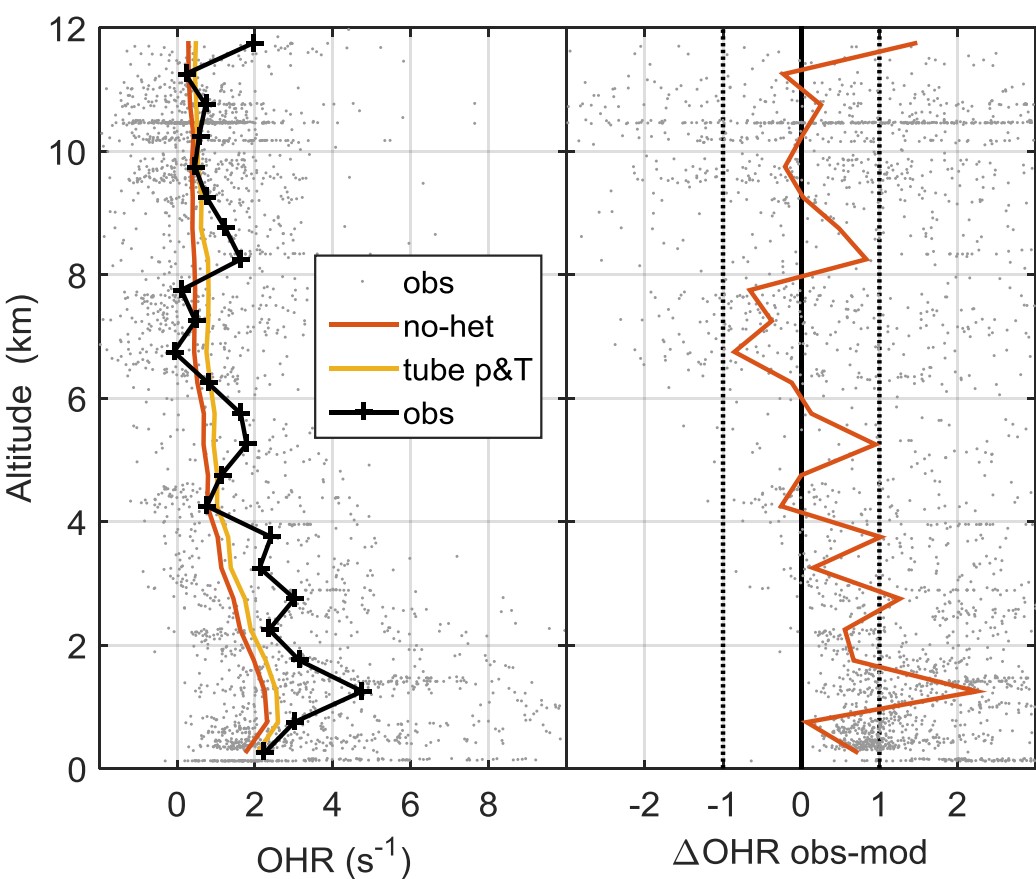

**Figure 5**. Median observed and model-calculated OH reactivity (left) and the difference between observed and modeled OH reactivity (right). The model-calculated OH reactivity was determined for ambient conditions (no-het) and for the OHR flow tube pressure and temperature (tube p&T). See text for explanation. Dotted vertical black lines on right are approximate indicators of measurement-model agreement.

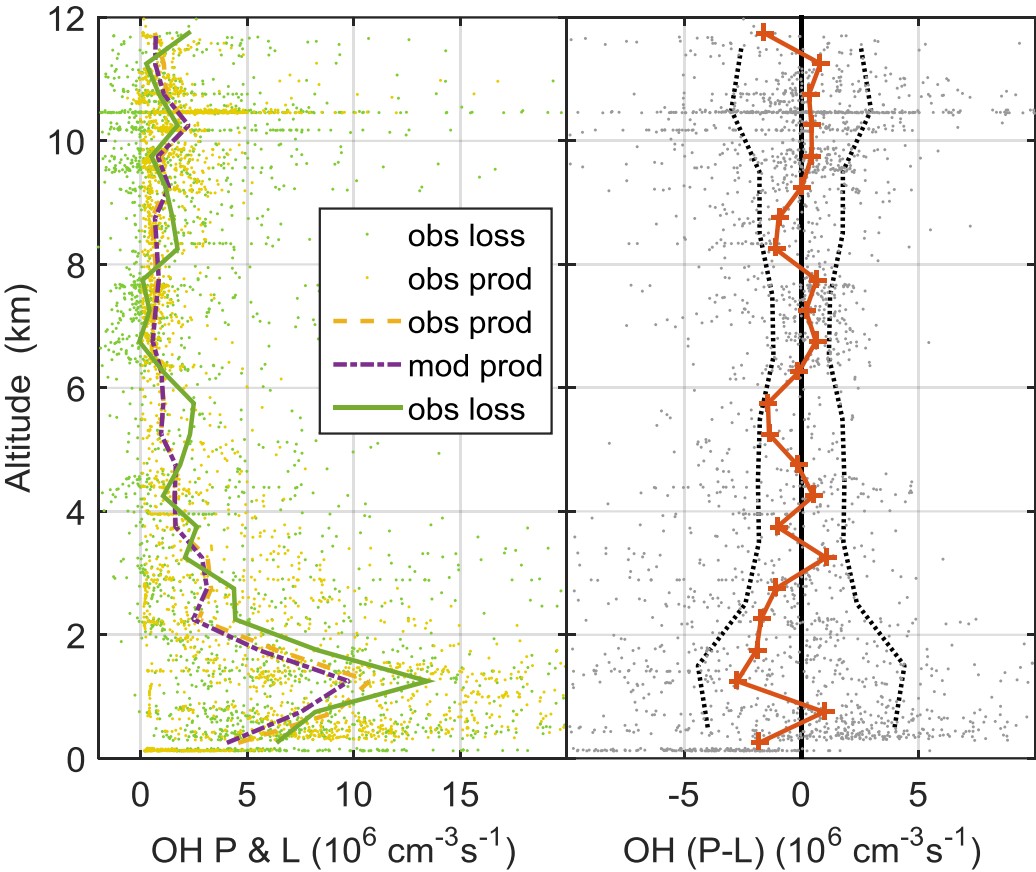

**Figure 6**. OH production and loss (left) and difference between measured OH production and measured OH loss, cm⁻³s⁻¹ divided (right). Modeled OH production and loss are equal. Dots are individual one-minute calculations of OH production using observed atmospheric constituents and photolysis frequencies. Dotted vertical black lines on right are indicators of OH production and loss agreement at the $1\sigma$ confidence and were determined by a propagation of error analysis.

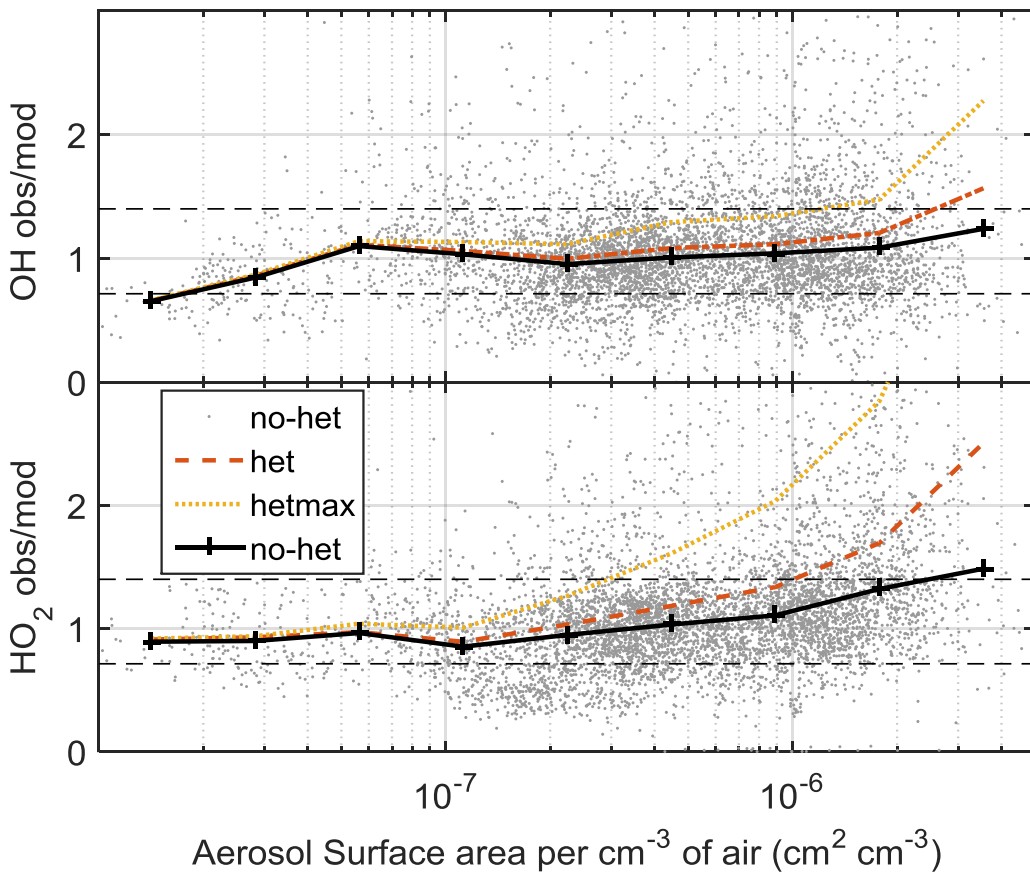

**Figure 7.** Median observed-to-modeled OH (top) and HO$_2$ (bottom) versus aerosol surface area per cm$^{-3}$ of air. Dotted horizontal black lines are approximate indicators of measurement-model agreement.

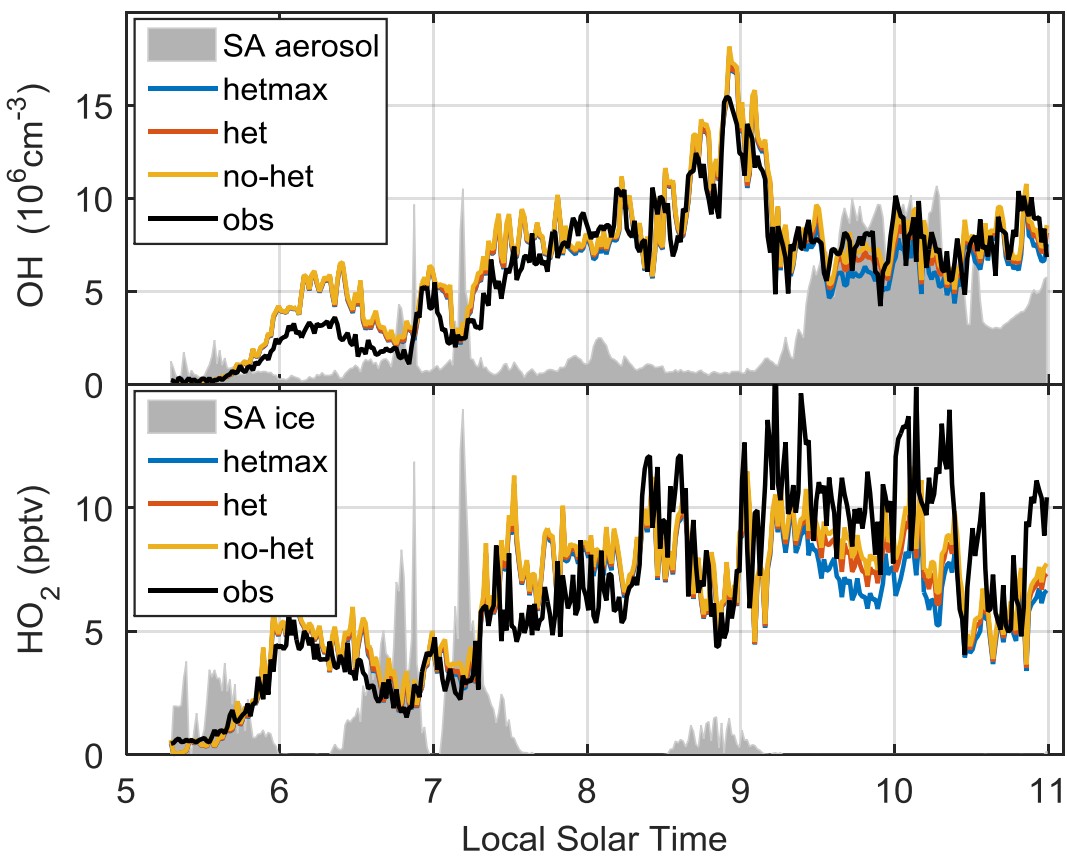

**Figure 8**. Evolution of OH (top) and HO₂ (bottom) downwind of a MCS. The het and hetmax models are essentially identical to the no-het model for this flight except in heavy aerosol amounts from 9 am to 11 am. The gray shading is scaled aerosol surface area (top panel) and is scaled ice surface area (bottom panel).

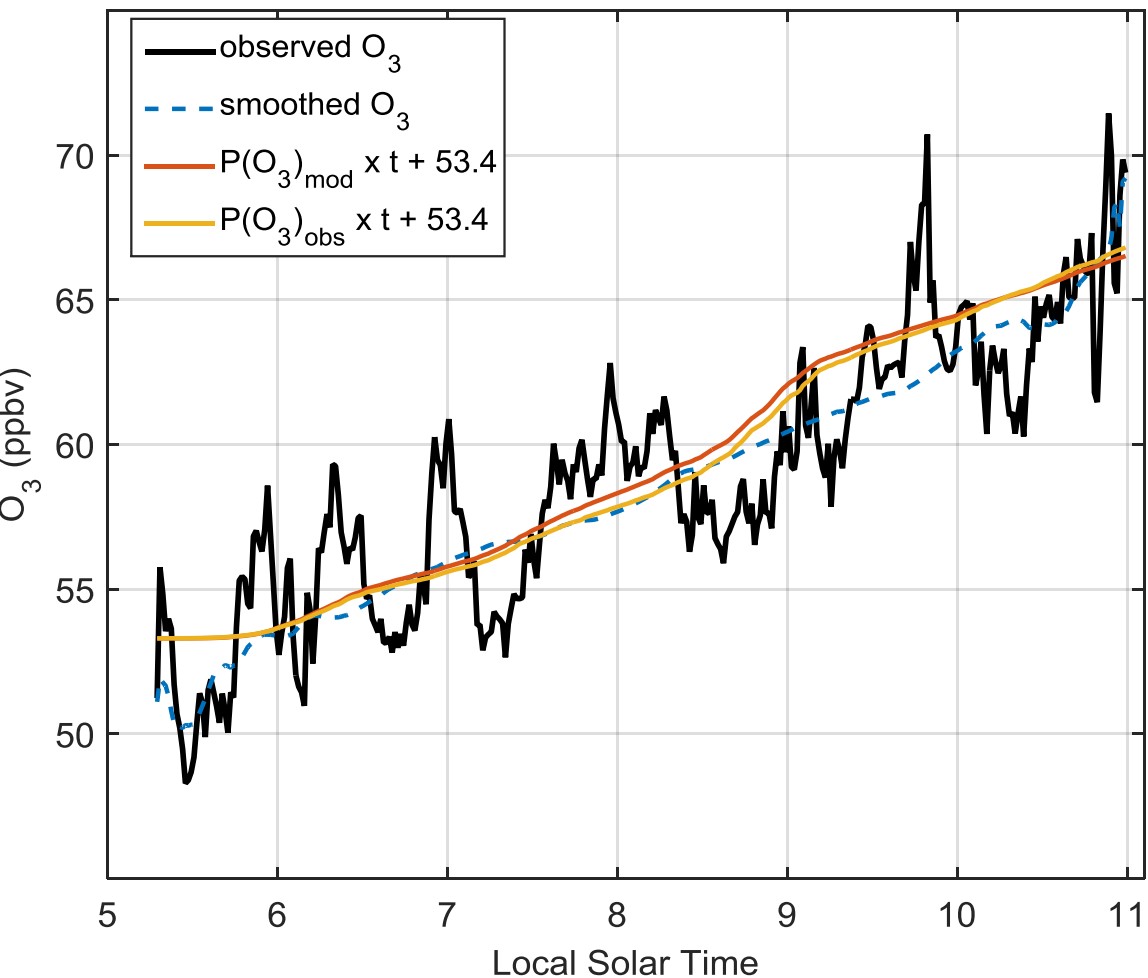

**Figure 9**. The $O_3$ trend in MCS outflow. Observed $O_3$ - one-minute data (black line) and smoothed data (dashed blue line, 180-min filter) - show variability along the legs set perpendicular to the air flow and also show the $O_3$ trend with time. The calculated $O_3$ uses the accumulated calculated $O_3$ production rates (P($O_3$) x time) for observed (yellow) and modeled $HO_2$ (red) and adds those values to an initial value of 53.4 ppbv to show the calculated $O_3$ trend.

Table 1. Measured chemical species

| Chemical species | Accuracy (2σ confidence) | Time resolution | Reference that describes the instrument |
|---|---|---|---|
| OH, $HO_2$ | ±32% | 20 s | Faloona et al., 2004 |
| NO; <br> $NO_2$; <br> $O_3$ | 0.01 ppbv + 4%; <br> 0.02 ppbv + 6%; <br> 0.04 ppbv + 3% | 1 s | Ryerson et al., 2000; <br> Pollack et al., 2011 |
| HCHO | ±10% + ±10 pptv offset | 1 s | Cazorla et al., 2015. |
| $NO_2$; <br> PNs; <br> MPN, ANs | ±5%; <br> ±10%; <br> ±15%; | 15 s | Thornton et al., 2000; <br> Day et al., 2002 |
| CO, $CH_4$, $N_2O$ | ±2% or 2 ppbv | 1 s | Sachse et al., 1991 |
| $SO_2$; <br> HCl; <br> PAN, PPN, $HNO_4$ | ±15% <br> ±20% <br> ±30% | 10 s | Huey, 2007 |
| $H_2O_2$, $CH_3OOH$, ISOPOOH, Glycolaldehyde, IEPOX, $C_5H_{10}O_3$, $C_5H_8O_3$, Ethanal_Nitrate, Hydroxyacetone, Hydrogen cyanide, $HNO_3$, Isoprene Nitrate, Peroxyacedic Acid, Propanone Nitrate | ±(40-80)% <br> ±50% + (25-100) pptv offset | 10 s | Crounse et al., 2006 |
| Isoprene, Monotepenes, MVK/MACR, Acetone/Propanal, Methanol, Acetaldehyde, Acetonitrile | ±10% | 2 s | Mielke, et al., 2008 |
| Ethyne, Ethane, Ethane, Propene, Propane, i-Butane, n-Butane, i-Pentane, n-Pentane, n-Hexane, n-Heptane, 2,3-Dimethylbutane, 2-Methylhexane, 3-Methylhexane, Cyclohexane, Benzene, Toluene, m,p-Xylene, o-Xylene, Ethylbenzene, cis-2-Butene, trans-2-Butene, C8-Aromatics/Benzaldehyde, DMS, Methyl nitrate, Isoprene, α-Pinene, β-Pinene, several halogen-containing compounds | ±10% | Variable, seconds to minutes | Colman et al., 2001 |
| $H_2O$ | ±5% or 1 ppmv | 1 s | Vay et al., 1998 |

**Table 2.** Scatter plot statistics for OH and $HO_2$

| case | molecule | units | slope | intercept | $R^2$ | ratio |
|---|---|---|---|---|---|---|
| no-het | OH | $10^6$ cm$^{-3}$ | 1.00 | 0.01 | 0.85 | 1.03 |
| | $HO_2$ | pptv | 1.00 | 0.36 | 0.68 | 1.10 |
| het | OH | $10^6$ cm$^{-3}$ | 1.07 | 0.02 | 0.87 | 1.11 |
| | $HO_2$ | pptv | 1.14 | 0.51 | 0.62 | 1.28 |
| hetmax | OH | $10^6$ cm$^{-3}$ | 1.24 | 0.03 | 0.85 | 1.29 |
| | $HO_2$ | pptv | 1.53 | 0.68 | 0.50 | 1.83 |