# Peer review of "Atmospheric Oxidation in the Presence of Clouds during the Deep Convective Clouds and Chemistry (DC3) Study"

_Atmospheric Chemistry and Physics, 2018_

## Referee Comment (RC1) · Anonymous Referee #2 · 20 Apr 2018

General comment

The manuscript describes airborne observations of OH, HO2 and OH reactivity during the DC3 project. The authors conclude that the generally good agreement between model predictions and the observations provides evidence that current photochemical models can simulate atmospheric oxidation even around clouds. Due to the large number of minor corrections that I came across during the review, however, along with a few more serious concerns, I remain to be convinced by the conclusions drawn. I hope the authors are able to adequately address the comments and corrections below before the final publication.

[Figure]

Major comments

The authors report short scattering signals by cloud particles which occur randomly both during online and offline measurements and discuss the treatment of the raw data to reduce the noise from these large scatter signals in section 2.2. Could the authors comment on whether this approach introduces any bias into the analysis, given that the online measurement period is 3 times longer than offline? The authors should provide a comment on the choice of exceeding the average background signal by four standard deviations. Would a different limit change the results?

On page 6, the OH and $HO_2$ interferences suffered by ATHOS are discussed and the authors state that interferences are only significant above forests and cities and are negligible outside of the planetary boundary layer. The authors should comment on whether this statement remains valid when sampling in and around convective clouds which can rapidly transport air from the surface. The correction applied to the $HO_2$ observations to account for $RO_2$ interferences is rudimentary and more details should be provided on the type of $RO_2$ species present. The correction will change with changing $RO_2$ species present and this relies on model predictions for $RO_2$ which is far from ideal.

Pg 12, section 3.3: Why was HONO photolysis not considered for OH production?

On page 14 the authors calculate $P(O_3)$ from the 'observed $HO_2$ + modelled $RO_2$' and from 'modelled $HO_2$ + modelled $RO_2$' and compare to the 'observed $O_3$ change'. Lines 16 and 17 state that $HO_2$ accounts for a little more than half the total $O_3$ production, but then lines 21 – 23 state that the observed $O_3$ change was equal to 14 ppbv and the $P(O_3)$ calculated from the observed $HO_2$ was 14 ppbv (figure 9 seems only to show $P(O_3)$ calculated using $HO_2$)? What was the contribution of modelled $RO_2$ to $P(O_3)$? Are the authors suggesting that roughly half of the ozone produced partitions to $NO_2$? Could the change in $NO_2$ over the flight leg be shown to corroborate this?

Minor corrections

Pg 2, lines 17 – 20: Reads as though nitric oxide is needed to produce RO2 and HO2 from OH oxidation of VOCs, this isn't the case.

Pg 3, lines 9 – 11: HO2 interferences in LIF systems can now be minimised, however, this sentence suggests that HO2 measurements are still affected by interferences. This needs to be corrected.

Pg 4, lines 1 – 2: I find this sentence confusing as metal ions enhance the HO2 uptake probability rather than reduce it.

Pg 5, line 15: 'lower' than what?

Pg 5, line 21: Provide the concentration of NO that is added.

Pg 9, line 14 & 15: Surface area is not a concentration. Please include the mean surface area.

Pg 9, line 24: Which organic peroxy radicals + OH reactions does the model include?

Pg 11, E3: k'OH should be defined absolutely. Does it just include OH + CO and OH + HCHO that lead to HO2 directly or are VOC reactions that cycle to HO2 via RO2 considered too? If it is the latter, the authors should comment on the validity of this approximation under low NO conditions.

Pg 11, line 20: Is this ratio from measurements or the model?

Pg 11, line 22: what is the uncertainty limit?

Pg 11, line 27: 'factor of 600 for JO(1D), a factor of 100 for NO.' it is not clear from these statements the range of photolysis rates or concentrations over which the model agrees with the observations. I suggest providing the range in the text.

Pg 11, line 28: 'One exception is OH for NO greater than 1 ppbv.' Are the observations greater or less than the model in this case?

Pg 12, line 3: Please provide the limit of detection here.

[Figure]

Pg 12, line 7: 'modeled OH reactivity looks quite similar..' Could % differences or differences in s-1 be given here?

Pg 12, line 8: what loss rate is assumed for the model generated intermediates?

Pg 13, line 22: 'substantial agreement' – could this be given as a % agreement?

Pg 13, line 29: provide figure number.

Pg 14, line 1: What was recalibrated in the instrument? The phototube? The authors should comment on why they think it is appropriate to apply a calibration performed so long after the campaign.

Pg 14, E3: should be E4. Also, please define L(O3) in the text.

Pg 16, line 1: '..HO2 calibration at low pressure was in error' wasn't a calibration over a range of pressures performed?

Pg 16, line 9: during which campaign was the fire plume observed?

Pg 16, line 21: add '3' after 'DC'

Pg 16, line 26 – 27: My understanding of the Whalley et al, 2015, mountaintop cloud study was that the uptake coefficient reported was determined from the observed decrease in the measured HO2 and was not calculated from the size, composition, etc of the aerosol/cloud droplet as is suggested in lines 26 and 27. The authors should compare the uptake coefficients determined rather than the level of decrease in HO2 given the very different surface areas encountered in the different campaigns.

Pg 16, line 28: 'small cloud impacts on HO2 are consistent..' this statement is at odds with the comment in the Introduction (pg 3, lines 19 – 22) which states that an observed to modelled ratio of 0.65 was reported during the in-cloud measurements in Olson et al (2006). Please clarify.

Pg 17, lines 9 – 11: was data missing when the model and measurements diverge?

Pg 17, line 23: remove 'strong'.

Figure 6: the dots representing 'obs loss' and 'obs prod' are very difficult to distinguish from each other. I suggest more contrasting colours are used. The modelled loss should be added to this figure too.

Figure S1 highlights the disagreement between OHRobs and OHRmod. If the modelled OHR was forced into agreement with the OHR observed, how would this impact the modelled OH (and the level of agreement with OH observed)?

Figure S1 – S6: The fitted line should be added to the OHR scatter plots.

Figure S8: is OH+NO2 considered as a loss?

Figure S9: could the temperature profile be added?

[Figure]

---

## Referee Comment (RC2) · Anonymous Referee #1 · 24 May 2018

General comments

The paper reports airborne observations of OH and $HO_2$ that were measured together with other trace gases and atmospheric parameters up to an altitude of 12 km during the DC3 mission over the central United States. The measured radical concentrations are compared to model calculations in order to test how well a photochemical box model can explain HOx in air close to deep convective clouds. In general, there is good agreement found between model results and measurements. Within the combined uncertainties, no significant difference is seen without and with heterogeneous chemistry using uptake coefficients for cloud particles consistent with laboratory findings. The data and the conclusions are convincing. The paper is well written and thematically suitable for publication in ACP. Before publication, the following comments should be addressed.

Major comments

1. Little if any information is provided on which organic compounds were measured during the flights. Were there other VOCs besides methane that made a significant contribution to the OH reactivity? Did isoprene play a role? This question is especially relevant in the outflow of convective clouds which can transport relatively short-lived species to higher altitudes in short time.

2. In previous publications about airborne HOx studies, in which some of the authors of this paper were involved, systematic NO-dependent deviations between measured and modelled $HO_2$ were reported (e.g., Ren et al., JGR 113, D05310, 2008; Olson et al., JGR 111, D10301, 2006). In the present study, no such deviations are found (Fig. S3). Can the authors comment on this different findings?

3. Besides their possible role in heterogeneous chemistry, cloud particles can influence HOx through their impact on solar UV radiation that is driving photolysis. Though this may not be in the focus of the present paper, have the authors tried to quantify the impact of clouds on j values influencing HOx? It could be interesting to see a with-cloud / without-cloud (clear sky) ratio of HOx as a function of solar zenith angle extracted (if possible) from the data set.

Minor comments

page 2, line19 - 20: the production of $RO_2$ does not require NO; please correct.

page 2, line 22: add a half sentence, why the production of NO and $O(^3P)$ produces new ozone.

page 2, line 23: add a short explanation, why in the absence of NO the formation of OH and $HO_2$ destroys ozone.

page 3, line17 and page 9, line 31: : typos; Kubistin.

page 6, lines 28 -32: how much is the limit of detection for OH and $HO_2$ affected by scattered light from cloud particles?

page 6, lines 23-25: the principle of the $k_{OH}$ measurement should be briefly explained in one or two sentences before equation (1) is presented. Alternatively, move equation (1) to page 7 between line 5 and 6.

page 7, line 23: the sentence is not clear. How large was the temperature difference between ambient air and air inside the flow tube?

page 8, line 6: The table showing the measurements is missing. It should be presented in the main paper. Accuracies and time resolution of the measurements should be given.

page 11, equation (3): how was k'OH calculated? Does it include VOCs (RH --> RO2 --> HO2), or only those species (e.g., CO, $O_3$, HCHO) that directly yield $HO_2$?

page 14, line 15: Fig. S8 ? References made in the main text to Figures S8, S9 and S10 should be checked and corrected.

---

## Author Comment (AC1) · 25 Jun 2018

Brune et al., Atmospheric Oxidation in the Presence of Clouds during the Deep Convective Clouds and Chemistry (DC3) Study, doi.org/10.5194/acp-2018-120

Response to Reviewer #1

We thank Reviewer #1 for useful comments. Below we answer the reviewer's concerns and make the necessary corrections to the paper and supplement. The reviewer's comments are in italics and our response in normal text.

*General comments*

*The paper reports airborne observations of OH and HO2 that were measured together with other trace gases and atmospheric parameters up to an altitude of 12 km during the DC3 mission over the central United States. The measured radical concentrations are compared to model calculations in order to test how well a photochemical box model can explain HOx in air close to deep convective clouds. In general, there is good agreement found between model results and measurements. Within the combined uncertainties, no significant difference is seen without and with heterogeneous chemistry using uptake coefficients for cloud particles consistent with laboratory findings. The data and the conclusions are convincing. The paper is well written and thematically suitable for publication in ACP.*

*Before publication, the following comments should be addressed.*

*Major comments*
*1. Little if any information is provided on which organic compounds were measured during the flights. Were there other VOCs besides methane that made a significant contribution to the OH reactivity? Did isoprene play a role? This question is especially relevant in the outflow of convective clouds which can transport relatively short-lived species to higher altitudes in short time.*

We provide the table the reviewer requested, but believe that it belongs in the supplemental material. The paragraph in Section 2.4 has been modified to be

"Accurate measurements of other chemical species and environmental variables are critical for this comparison of measured and modeled OH, $HO_2$, and OH reactivity. The photolysis frequency measurements are particularly critical for DC3 because of all the time spent flying around clouds and in the deep convection anvil. A list of these measurements is given in Table S1 and is summarized in Barth et al. (2015; 2016) and Pollack et al. (2016). The list of measured chemical species includes CO, $CH_4$, $N_2O$, NO, $NO_2$, $O_3$, organic nitrates, alkanes, alkenes, aromatics, aldehydes, alcohols, and peroxides."

We also add a paragraph to the end of section 3.2 that reads

"According to the model calculations, CO contributes the most to the OH reactivity, with ~20% in the PBL and 30-40% aloft. Next is $CH_4$ at (5-10)%, HCHO at (5-10)%, $O_3$ at (2-10)%, and $CH_3CHO$ at ~5%. Isoprene was typically (1-2)%, except in some PBL plumes where it was as much as 60% and in some anvils where it was as much as 20% (0.1-0.5 $s^{-1}$). The most significant ten chemical species were all measured and account for (60-70)% of the total model-calculated OH reactivity."

*2. In previous publications about airborne HOx studies, in which some of the authors of this paper were involved, systematic NO-dependent deviations between measured and modelled HO2 were reported (e.g., Ren et al., JGR 113, D05310, 2008; Olson et al., JGR 111, D10301, 2006). In the present study, no such deviations are found (Fig. S3). Can the authors comment on this different findings?*

We thank the reviewer for this observation. Figures 4 and (now) S4 show averages over NO bins, including all flights and altitude. If we plot one-minute $HO_2$ observed-to-modeled ratios for DC3 flights, the results are quite similar to those in Faloona et al. (1999) when re-evaluated by Olson et al. (2006). The substantial deviations observed during ground-based studies typically are not evident until NO exceeds 3 ppbv, which is the upper limit of NO for the data used from this DC3 study.

In the last paragraph in section 3.1, we have thus added the following text:
"The observed-to-modeled $HO_2$ ratio shows little evidence of a NO-dependence. The reanalysis of Olson et al. (2006) explains the NO-dependence of the ratio discussed in Faloona et al., (1999), and the NO-dependence observed often in ground-based studies is not evident until NO exceeds 3-5 ppbv. Because the highest NO mixing ratio encountered in DC3 was about 3 ppbv, we would not expect to see this effect in the DC3 results."

*3. Besides their possible role in heterogeneous chemistry, cloud particles can influence HOx through their impact on solar UV radiation that is driving photolysis. Though this may not be in the focus of the present paper, have the authors tried to quantify the impact of clouds on j values influencing HOx? It could be interesting to see a with-cloud / without-cloud (clear sky) ratio of HOx  as a function of solar zenith angle extracted (if possible) from the data set.*

We agree that this is an interesting study to do, but it not within the scope of this paper. However, we did a quick check on this idea by calculating the observed-to-modeled $HO_2$ ratio, filtering it for altitude bins (i.e., 5-8 km, 8-10 km, >10 km), and solar zenith angle bins (i.e., 50°-60°, 60°-70°, 70°-80°), and then plotting it as a function of $J(O(^1D))$. In all cases, the ratio was constant as a function of $J(O(^1D))$. This observation indicates that using the measured photolysis frequencies accurately captures the $HO_x$ production by photolysis as seen in Figure 4e.

*Minor comments*
*page 2, line19 - 20: the production of RO2 does not require NO; please correct.*

Corrected.

*page 2, line 22: add a half sentence, why the production of NO and O(3 P) produces new ozone.*

The end of that sentence was rewritten as "…which then absorbs solar radiation to produce NO and $O(^3P)$, and $O(_3P)$ combines with $O_2$ to form new $O_3$."

*page 2, line 23: add a short explanation, why in the absence of NO the formation of OH and HO2 destroys ozone.*

The end of that sentence as rewritten as "… acts to destroy ozone through OH production and the reactions $O_3+HO_2$ and $O_3+OH$."

*page 3, line17 and page 9, line 31: : typos; Kubistin.*

Fixed.

*page 6, lines 28 -32: how much is the limit of detection for OH and HO2 affected by scattered light from cloud particles?*

In Section 2, 2$^{nd}$ paragraph, we add the statement "… which reduces the measurement precision by as much as a factor of five."

*page 6, lines 23-25: the principle of the kOH measurement should be briefly explained in one or two sentences before equation (1) is presented. Alternatively, move equation (1) to page 7 between line 5 and 6.*

We added two sentences in the first paragraph in Section 2.3:

"It is directly measured by adding OH to the air flowing through a tube and then monitoring the decay of the logarithm of the OH signal as the reaction time between the OH addition and OH detection is increased. OH can also be lost to the tube walls, so the measured OH reactivity must be corrected for this wall loss."

*page 7, line 23: the sentence is not clear. How large was the temperature difference between ambient air and air inside the flow tube?*

This sentence has been rewritten as the following:
"Air entering the OHR flow tube was warmed by ~5$^o$C at altitudes below 2 km to by as much as 75$^o$C at 12 km."

*page 8, line 6: The table showing the measurements is missing. It should be presented in the main paper. Accuracies and time resolution of the measurements should be given.*

We have included Table S1 in the supplement, where we think it belongs.

*page 11, equation (3): how was k'OH calculated? Does it include VOCs (RH --> RO2 --> HO2), or only those species (e.g., CO, O3, HCHO) that directly yield HO2?*

We have removed this equation because the total OH reactivity calculated from the model was found by summing all the modeled loss rates and then dividing this number by the OH concentration for each time step. Every chemical species that was used to constrain the model or that was created by the model was used to find the model-calculated OH reactivity.

In Section 2.5, we revised the 7$^{th}$ paragraph on the modeled OH reactivity to be
"OH reactivity was also modeled for comparison to observed OH reactivity. Modeled OH reactivity was calculated from the measured chemical species plus OH reactants that were not measured but were produced by the photochemical model. Examples of these additional OH reactants are organic peroxy, organic peroxides, and unmeasured aldehydes. The OH reactivity calculated from only the measured chemical species accounted for more than 90% of the total OH reactivity. Uncertainty in the modeled OH reactivity is estimated to be ±10%, 1$\sigma$ confidence (Kovacs et al., 2001)"

*page 14, line 15: Fig. S8 ? References made in the main text to Figures S8, S9 and S10 should be checked and corrected*

Corrected

Brune et al., Atmospheric Oxidation in the Presence of Clouds during the Deep Convective Clouds and Chemistry (DC3) Study, doi.org/10.5194/acp-2018-120

We thank Reviewer #2 for a careful review of this paper. Below we answer the reviewer's concerns and make the necessary corrections to the paper and supplement.

A possible concern of the reviewers is that we did not properly limit the conclusion of generally good agreement between model and measurement to the DC3 study. As the reviewer as pointed out, there are elements of the model that are quite uncertain. Therefore, the agreement between observed and modeled OH and $HO_2$ to within combined uncertainties suggests that, at this level of uncertainty, the atmospheric oxidation chemistry is understood. However, if the uncertainties could be reduced a factor of two or more, then this comparison could possibly uncover omissions or errors in the model chemical mechanism. We hope that Reviewer #2 is convinced of these statements.

We have modified the end of the abstract to read:

"Even with some observed-to-modeled discrepancies, these results provide evidence that a current measurement-constrained photochemical model can simulate observed atmospheric oxidation processes to within combined uncertainties, even around convective clouds. For this DC3 study, reduction in the combined uncertainties would be needed to confidently unmask errors or omissions in the model chemical mechanism."

We have also rewritten the final paragraph in the Conclusion:

"Even with these observed-to-modeled discrepancies, the general agreement for observed and modeled OH and $HO_2$ suggests that current photochemical models can simulate observed atmospheric oxidation processes even around clouds to within these combined uncertainties. Reducing these uncertainties will enable comparisons of observed and modeled OH and $HO_2$ to provide a more stringent test of the understanding of atmospheric oxidation chemistry and thus to lead to an improvement in that understanding."

We note that the photolysis frequencies were revised just after we submitted the manuscript. The changes in OH and $HO_2$ mixing ratios are small. Using the corrected photolysis frequencies, both OH and $HO_2$ are on average 2% larger, with only 5% of results giving differences greater than 15%, primarily at low solar zenith angles. On the figures in the manuscript, the differences are hardly perceptible. The merge file with the corrected photolysis frequencies was used for all the modeling done for this manuscript.

*Major comments.*

*The authors report short scattering signals by cloud particles which occur randomly both during online and offline measurements and discuss the treatment of the raw data to reduce the noise from these large scatter signals in section 2.2. Could the authors comment on whether this approach introduces any bias into the analysis, given that the online measurement period is 3 times longer than offline? The authors should provide a comment on the choice of exceeding the average background signal by four standard deviations. Would a different limit change the results?*

This approach is taken in order to maximize the ambient measurements of OH and $HO_2$, allowing the possibility of higher frequency measurements, especially for $HO_2$, without sacrificing any precision. The on-line and off-line signals are summed and then averaged, and because the averages are subtracted, this method introduces no bias. However, the signal-tonoise ratio is actually better for the 15-5 versus the 10-10 configuration. For OH, the daytime on-line signal is typically 10 cts s$^{-1}$ while the off-line signal is typically 1 ct s$^{-1}$. For 15-5, S/N is approximately 145/sqrt(150+5) = 11 and for 10-10, S/N is approximately 90/sqrt(100+10) = 9. For HO$_2$, the daytime on-line signal is typically 100 cts s$^{-1}$ and the off-line signal is typically 1 ct s$^{-1}$, so that both S/N and data coverage are improved by using 15 s on and 5 s off.

The large scattering signals due to cloud particles can be as much as 100's of cts s$^{-1}$, but they occur randomly in any second. So averaging the on-line and off-line signals and then taking the difference does not introduce any bias. However, the difference signal gets rather noisy because cloud spikes can occur randomly and unevenly between on-line and off-line segments. We chose 4 standard deviations, but got the same results for OH and HO$_2$ to within 3% for OH and 4% for HO$_2$ for using the range of 2 to 6 standard deviations.

At the end of the 2$^{nd}$ paragraph in Secton 2.2, we added

"Less than 3% of the data were removed. The overall results for OH and HO$_2$ vary less than 4% for filtering between two and six standard deviations."

*On page 6, the OH and HO2 interferences suffered by ATHOS are discussed and the authors state that interferences are only significant above forests and cities and are negligible outside of the planetary boundary layer. The authors should comment on whether this statement remains valid when sampling in and around convective clouds which can rapidly transport air from the surface. The correction applied to the HO2 observations to account for RO2 interferences is rudimentary and more details should be provided on the type of RO2 species present. The correction will change with changing RO2 species present and this relies on model predictions for RO2 which is far from ideal.*

We agree that relying on the model to correct the HO$_2$ measurements is far from ideal. But while we have now had this capability for 5 years, we did not have it properly implemented for DC3 in summer 2012. However, as shown below, the effects are not large for most of the DC3 HO$_2$ values.

We have completely redone how we do the correction for the RO$_2$ interference in the HO$_2$ measurement. It now uses MCMv3.2.1 modeled RO$_2$ species inside the ATHOS detection flow tube. Knowing the flow through ATHOS, we calculate the concentration of the reagent NO, which was about 3x10$^{13}$ cm$^{-3}$. The reaction time was determined to be 3.7 ms, as verified by the HO$_2$ conversion rate measured in the laboratory. These values were put into the model and the RO$_2$-produced HO$_2$ interference was calculated for each one-minute time step. Ambient HO$_2$ was then determined by subtracting this modeled HO$_2$ interference from the measured HO$_2$. The HO$_2$ correction is now slightly different from the previous correction at all altitudes.

Sec5tion 2.2, 4$^{th}$ paragraph. We revise the text as

"For HO$_2$, the correction method uses more than 1000 RO$_2$ chemical species modeled by MCMv3.3.1 and assumes that they are ingested into the detection flow tube without any wall loss. The model then calculates the resulting OH, which is what would be detected as HO$_2$. The calculated concentration of reactant NO is ~3x10$^{13}$ cm$^{-3}$ and the reaction time was determined to be 3.7 ms, as verified by the HO$_2$ conversion rate measured in the laboratory. This calculation was repeated for each one-minute time step and this calculated interference was then subtracted from the observed HO$_2$, resulting in the HO$_2$ values reported here. Observed HO$_2$ was reduced by an average of 2%, with some peaks, both in the PBL and aloft, where it was 10%. Because the model RO$_2$ mechanisms are uncertain, the uncertainty for this correction is estimated by be a factor of 2, which increases the absolute uncertainty for HO$_2$ from ±16% to ±20%, 1σ confidence."

*Pg 12, section 3.3: Why was HONO photolysis not considered for OH production?*

We have added OH production by HONO photolysis and OH loss by OH+NO+M to Figure S11. The reason we did not include them before is that the OH production by HONO almost exactly balances the OH loss by HONO formation at all altitudes, which implies that surface HONO production is not important for DC3, even in the PBL. Because there is so little HONO down low, the HONO above 8 km is likely formed from *in situ* OH reactions with lightning or stratospheric NO. Thus HONO seems to be acting more as a $HO_x$ reservoir than a $HO_x$ source.

We add the following sentence to the last paragraph of section 4.1:

"$HO_x$ production by HONO photolysis essentially balances $HO_x$ loss by HONO formation."

*On page 14 the authors calculate P(O3) from the 'observed HO2 + modelled RO2' and from 'modelled HO2 + modelled RO2' and compare to the 'observed O3 change'. Lines 16 and 17 state that HO2 accounts for a little more than half the total O3 production, but then lines 21 – 23 state that the observed O3 change was equal to 14 ppbv and the P(O3) calculated from the observed HO2 was 14 ppbv (figure 9 seems only to show P(O3) calculated using HO2)? What was the contribution of modelled RO2 to P(O3)? Are the authors suggesting that roughly half of the ozone produced partitions to NO2? Could the change in NO2 over the flight leg be shown to corroborate this?*

We assumed erroneously that the reader would know that we added the modeled RO2 to the observed or modeled HO2 to calculate the ozone production rate. The modeled RO2 is always included in the calculated P(O3) for the two cases, one using modeled HO2 and the other using measured HO2. In no case do we determine $P(O_3)$ from $HO_2$ alone. We make this clear by adding the words "and modeled RO2" to every figure and every case that we talk about $P(O_3)$.

Section 3.5, 4th paragraph, we also add the following sentence to make it clear that these two $RO_2$ chemical species are the most important in this calculation:

"Modeled $RO_2$ is primarily $CH_3O_2$ and $CH_3CHO_2$ above 5 km."

*Minor corrections*

*Pg 2, lines 17 – 20: Reads as though nitric oxide is needed to produce RO2 and HO2 from OH oxidation of VOCs, this isn't the case.*

Thanks for pointing out this poor wording. We have corrected this wording, removing "and in the presence of nitrogen oxides," and replaced it with "including"

*Pg 3, lines 9 – 11: HO2 interferences in LIF systems can now be minimised, however, this sentence suggests that HO2 measurements are still affected by interferences. This needs to be corrected.*

Corrected. Please see above in major comments.

*Pg 4, lines 1 – 2: I find this sentence confusing as metal ions enhance the HO2 uptake probability rather than reduce it.*

We have rewritten the 7[th] paragraph in the Introduction as

"Laboratory studies show that the $HO_2$ effective uptake coefficient on moist aerosol particles that contain copper is probably much less than 0.01 in the lower troposphere but may be greater than 0.1 in the upper troposphere (Thornton et al., 2008). However, other laboratory studies show that adding organics to the particles or lowering the relative humidity can reduce uptake coefficients (Lakey et al., 2015; Lakey et al., 2016). These values are generally lower than the $HO_2$ effective uptake coefficient assumed in global chemical transport models."

*Pg 5, line 15: 'lower' than what?*

Corrected. It should be "low".

*Pg 5, line 21: Provide the concentration of NO that is added.*

We add this in the last paragraph in section 2.2. It was $3 \times 10^{13}$ cm$^{-3}$.

*Pg 9, line 14 & 15: Surface area is not a concentration. Please include the mean surface area.*

The units show that it is not a surface area either, which would be simply cm$^2$. It is a mean surface area per cm$^{-3}$ of air, as is used in Seinfeld and Pandis. We will make this change throughout the manuscript. Also, it is not the mean surface area per cm$^{-3}$ of air, but instead is the total surface area per cm$^{-3}$ of air. We do find the mean particle radius though.

*Pg 9, line 24: Which organic peroxy radicals + OH reactions does the model include?*

We suspect that the reviewer knows very well that MCMv3.3.1 does not include the $RO_2$+OH reaction. When we include this reaction for $CH_3O_2$ and $C_2H_5O_2$, as in Assaf et al., (EnvSci.Tech, 51, 2017), we get results that are a few percent different from not including these reactions. We have added a paragraph at the end of section 2.5 to describe how we did this modeling:

"The model was run 27 times to test the sensitivity of the calculated OH and $HO_2$ to different factors. First, the chemical mechanism was expanded to include the reactions of $CH_3O_2$+OH and $C_2H_5O_2$+OH (Assaf et al., 2017), and in some cases, reactions of OH with the next 300 most significant RO2 species that comprise 95% of the modeled RO2 total, assuming a reaction rate coefficient of 10-10 cm-3s-1.  Adding the measured reactions decreased OH by ~1% (5% maximum) and increased $HO_2$ by <1% (15% maximum); adding the additional the assumed reactions changed modeled OH and $HO_2$ by less than 3%. Second, the decay time of the unconstrained modeled oxygenated intermediates was varied from 6 hours to 5 days. The resulting modeled OH and $HO_2$ varied by ~10% over this range. This decay time serves as a proxy for surface deposition in the planetary boundary layer as well as for recently measured rapid reactions of highly oxidized $RO_2$ + $RO_2$ to form peroxides (Bernt et al., 2018). This decay time is highly uncertain. The mean of the model runs using decay times of 6 hours, 1 day, and 5 days and using the model mechanism including $CH_3O_2$+OH and $C_2H_5O_2$+OH was used for the comparisons to the measurements."

*Pg 11, E3: k'OH should be defined absolutely. Does it just include OH + CO and OH + HCHO that lead to HO2 directly or are VOC reactions that cycle to HO2 via RO2 considered too? If it is the latter, the authors should comment on the validity of this approximation under low NO conditions.*

The reviewer has pointed out the well-known difficulties with this simple relationship. We do not use this approximation but wanted to include this expression in the paper to illustrate that the ratio is increased when the OH reactivity is higher and decreased when NO is higher. We remove it.

*Pg 11, line 20: Is this ratio from measurements or the model?*

We clarify this by changing this sentence to the following:
"Both the observed and modelled $HO_2$/OH ratios are greater than 100 below 4 km and fall to less than 10 at 12 km (Fig. S2)."

*Pg 11, line 22: what is the uncertainty limit?*

We add the following:
"… ($1\sigma$ confidence) of a factor of 1/1.4 to 1.4."

Using this uncertainty limit is valid because we estimate that about half the uncertainty for OH and $HO_2$ comes from factors that are common to both calibrations while the other approximate half comes from factors that are specific to each measurement. For instance, both are affected the same by the uncertainty in the lamp flux of the calibration system, but their uncertainties related to laser power monitoring in the two different axes and to developing a fit to the calibration data are unique to the OH and $HO_2$ measurements.

*Pg 11, line 27: 'factor of 600 for JO(1D), a factor of 100 for NO..' it is not clear from these statements the range of photolysis rates or concentrations over which the model agrees with the observations. I suggest providing the range in the text.*

We have modified the text as recommended to read the following:
"In general, measured and model OH and $HO_2$ agree from $2 \times 10^{-6}$ $s^{-1}$ to $7 \times 10^{-5}$ $s^{-1}$ for JO($^1$D), from $2 \times 10^{-3}$ ppbv to $7 \times 10^{-1}$ ppbv for NO, from 40 ppbv to 100 ppbv for $O_3$."

*Pg 11, line 28: 'One exception is OH for NO greater than 1 ppbv..' Are the observations greater or less than the model in this case?*

This sentence has been removed as it is not essential.

*Pg 12, line 3: Please provide the limit of detection here.*
We add the sentence:
"for 20-second measurements is estimated to be about 0.6 $s^{-1}$, which means that most OH reactivity measurements were at or below the limit-of-detection."

*Pg 12, line 7: 'modeled OH reactivity looks quite similar..' Could % differences or differences in s-1 be given here?*

We change this sentence to the following:
" … but for the entire altitude, the difference between the observed and modeled OH reactivity is 0.2 $s^{-1}$ larger for the modelled OH reactivity at OHR flow tube conditions than for the modelled OH reactivity at ambient conditions, a difference that is swamped by the noise in the observed OH reactivity."

*Pg 12, line 8: what loss rate is assumed for the model generated intermediates?*

In the free troposphere, the deposition loss of model-generated intermediates is probably several days, since their loss rate depends on their deposition. Some may be taken up in cloud drops, in which case their loss rate would be much faster. We do not know for certain which case is true. Therefore, the modeled values we report come from an ensemble of models, the decay frequencies are varied from $2 \times 10^{-6}$ $s^{-1}$ (6 hours) to $5 \times 10^{-5}$ $s^{-1}$ (5 days). How we handled this in the modeling is described in the new last paragraph in section 2.5, Photochemical Box Model.

*Pg 13, line 22: 'substantial agreement' – could this be given as a % agreement?*

We have rewritten this paragraph as the following:

"In Figures 2-4 and S6-S7 and Tables 1 and S1, the no-het and het models agree with the observed OH and $HO_2$ to within the combined uncertainties, with the exception of the het model for $HO_2$ when the aerosol surface area per cm-3 of air was greater than $10^{-6}$ $cm^2$ $cm^{-3}$. In that case, the ratio of the observed-to-modeled $HO_2$ was too large, indicating that the het model was reducing $HO_2$ too much. On the other hand, the difference between the observations and hetmax model is greater than the uncertainty limits in almost every comparison. When the OH observed-to-modeled ratio is plotted as a function of aerosol surface area (Fig. 7), the no-het model gives a better agreement with the observations as a function of surface area than the het model does. For $HO_2$, the difference between the het model and the observations exceeds the combined uncertainty limits when the aerosol surface area per $cm^{-3}$ of air exceeds $10^{-6}$ $cm^2cm^{-3}$.

*Pg 13, line 29: provide figure number.*

Done.

*Pg 14, line 1: What was recalibrated in the instrument? The phototube? The authors should comment on why they think it is appropriate to apply a calibration performed so long after the campaign.*

Looking back through the notes for DC3, we noticed that the procedure had not been followed completely and that the transmission of the window between the calibration lamp and the calibration flow tube, which is 0.86, had not been included in the calibration. We re-measured the window transmission and confirmed that it was still 0.86 and then applied this number to the OH and $HO_2$ calibration.

*Pg 14, E3: should be E4. Also, please define L(O3) in the text.*

Since we removed Equation 3, this equation is now E3 again.

*Pg 16, line 1: '..HO2 calibration at low pressure was in error' wasn't a calibration over a range of pressures performed?*

For ATHOS, OH and $HO_2$ are always calibrated over a range in pressure that extends from a lower pressure than encountered in flight to a higher pressure than encountered in flight. For our instrument, the calibration has the shape similar to a Plank distribution of spectral irradiance – it rises steeply on the low pressure and then decreases more slowly on the high pressure end.

Thus, the lower pressure end is the most difficult to calibrate because small errors of even 5% in the calibrations numbers can lead to large errors in the fitted calibration curve.

*Pg 16, line 9: during which campaign was the fire plume observed?*
The fire plume was encountered on 22 June, day-of-year 174. In Figure S1, the encounter took place near the end of doy 174, when the DC-8 dipped down in altitude. The SZA was well above 85°. so OH was quite low.

We modified this sentence to read
"A spike in excess of 5 ppbv hr$^{-1}$ was found in a fire plume on 22 June (day-of-year 174), where abundances of both VOCs and NO$_x$ were enhanced."

*Pg 16, line 21: add '3' after 'DC'*

Fixed

*Pg 16, line 26 – 27: My understanding of the Whalley et al, 2015, mountaintop cloud study was that the uptake coefficient reported was determined from the observed decrease in the measured HO2 and was not calculated from the size, composition, etc of the aerosol/cloud droplet as is suggested in lines 26 and 27. The authors should compare the uptake coefficients determined rather than the level of decrease in HO2 given the very different surface areas encountered in the different campaigns.*

We have fixed this issue, as shown in the answer to the next reviewer comment.

*Pg 16, line 28: 'small cloud impacts on HO2 are consistent..' this statement is at odds with the comment in the Introduction (pg 3, lines 19 – 22) which states that an observed to modelled ratio of 0.65 was reported during the in-cloud measurements in Olson et al (2006). Please clarify.*

We are now clearer on the distinction between measurements of HO$_2$ uptake on ice particles versus liquid cloud particles.

We have rewritten Section 4.2, 2$^{nd}$ paragraph as
"In DC3, the DC-8 spent hours flying in anvils of the cumulus clouds, which consisted of ice particles.  DC3 provides evidence that the HO$_2$ uptake on ice is small. These results are consistent with HO$_2$ results over the western Pacific Ocean (Olson et al., 2004) but not with those over the northern Atlantic (Jeaglé et al., 2000). In Mauldin et al. (1998), a large difference between the observed and modeled OH was found in clouds, but this difference may have been due to the lack of photolysis frequency measurements, which are crucial to test photochemistry in a cloudy environment. In DC3, the DC-8 spent essentially no time in liquid clouds, for which there is evidence of measurable HO$_2$ uptake (Olson et al., 2006; Whalley et al., 2015). Thus these DC3 results provide constraints of HO$_2$ uptake on aerosol and ice particles, but not on liquid water particles."

*Pg 17, lines 9 – 11: was data missing when the model and measurements diverge?*

This is a good point. The answer is "no". The 10% of the time that all the constraining measurements are simultaneous show the same amount of agreement and discrepancy as the entire data set.

*Pg 17, line 23: remove 'strong'.*

Removed in the rewrite of this paragraph.

*Figure 6: the dots representing 'obs loss' and 'obs prod' are very difficult to distinguish from each other. I suggest more contrasting colours are used. The modelled loss should be added to this figure too.*

Done. Model loss is identical to model production, so to the figure caption we have added the sentence
"Model OH production and loss are equal."

*Figure S1 highlights the disagreement between OHRobs and OHRmod. If the modelled OHR was forced into agreement with the OHR observed, how would this impact the modelled OH (and the level of agreement with OH observed)?*

For DC3, the observed OH reactivity is significantly greater than the limit-of-detection only below 2 km altitude. The largest plumes, with OH reactivity exceeding 5 s$^{-1}$, are generally over forests, where the OH measurement is known to suffer from an interference. Not that much time was spent flying over forests. So for these DC3 results, we do not think that this exercise would be instructive.

*Figure S1 – S6: The fitted line should be added to the OHR scatter plots.*

This fit is pretty useless because the slope is determined by the noise in the observed OH reactivity with values below.

*Figure S8: is OH+NO2 considered as a loss?*

Yes it is.

*Figure S9: could the temperature profile be added?*

Done.

[revised manuscript text omitted]

|---|---|---|---|
| OH, HO$_2$ | ±32% | 20 s | Faloona et al., 2004 |
| NO;
NO$_2$;
O$_3$ | 0.01 ppbv + 4%; 0.02 ppbv + 6%; 0.04 ppbv + 3% | 1 s | Ryerson et al., 2000;
Pollack et al., 2011 |
| HCHO | ±10% + ±10 pptv offset | 1 s | Cazorla et al., 2015. |
| NO$_2$;
PNs;
MPN, ANs | ±5%;
±10%;
±15%; | 15 s | Thornton et al., 2000;
Day et al., 2002 |
| CO, CH$_4$, N$_2$O | ±2% or 2 ppbv | 1 s | Sachse et al., 1991 |
| SO$_2$;
HCl;
PAN, PPN, HNO$_4$ | ±15%
±20%
±30% | 10 s | Huey, 2007 |
| H$_2$O$_2$, CH$_3$OOH, ISOPOOH, Glycolaldehyde, IEPOX, C$_5$H$_{10}$O$_3$, C$_5$H$_8$O$_3$, Ethanal_Nitrate, Hydroxyacetone, Hydrogen cyanide, HNO$_3$, Isoprene nitrate, Peroxyacedic Acid, Propanone Nitrate | ±(40-80)%
±50% + (25-100) pptv offset | 10 s | Crounse et al., 2006 |
| Isoprene, Monotepenes, MVK/MACR, Acetone/Propanal, Methanol, Acetaldehyde, Acetonitrile | ±10% | 2 s | Mielke, et al., 2008 |
| Ethyne, Ethane, Ethane, Propene, Propane, i-Butane, n-Butane, i-Pentane, n-Pentane, n-Hexane, n-Heptane, 2,3-Dimethylbutane, 2-Methylhexane, 3-Methylhexane, Cyclohexane, Benzene, Toluene, m,p-Xylene, o-Xylene, Ethylbenzene, cis-2-Butene, trans-2-Butene, C8-Aromatics/Benzaldehyde, DMS, Methyl nitrate, Isoprene, α-Pinene, β-Pinene, several halogen-containing compounds | ±10% | Variable, seconds to minutes | Colman et al., 2001 |
| H$_2$O | ±5% or 1 ppmv | 1 s | Vay et al., 1998 |

[Figure]

**Figure S1.** Comparison of observed and modeled OH and HO$_2$ as a function of time. Blue lines are observations. Models with different integration times, dilution times, and with and without added RO$_2$+OH chemistry are plotted

in different colors to show the range of modeled OH and $HO_2$ values for these different model scenarios. Scaled altitude (black line) and separation between flights (vertical dotted lines) are added for clarify. If the observed blue line cannot be seen, then observed and modeled values agree.

[Figure]

[Figure]

**Figure S21**. Observations versus the no-het model. Scatter plots of observed versus modeled OH (upper right), HO₂ (upper left), HO₂/OH (lower left), and OH reactivity (lower right). Gray points are one-minute averages. Dashed red lines are factors of 1/1.4 and 1.4 times the fitted line (solid red).

[Figure]

[Figure]

[Figure]

**Figure S3**. Median measured and modeled HO$_2$/OH as a function of altitude (left); median ratio of observed-to-modeled HO$_2$/OH for the three models as a function of altitude. Gray points are individual 1-minute HO$_2$/OH observations and ratios of observed-to-no-heterogeneous modeled HO$_2$/OH. Dotted vertical red lines on right are approximate indicators of observation and model agreement.

[Figure]

[Figure]

**Figure S43**. Measured/modeled OH (left) and HO₂ (right) as a function of controlling variables: JO($^1$D) in s$^{-1}$, NO in ppbv, O₃ in ppbv, and modeled OH reactivity in s$^{-1}$. Ratios are averaged for three different altitude bands and all the data. Dotted horizontal lines are approximate indicators of observation and model agreement.

[Figure]

[Figure]

[Figure]

**Figure S54.** Median observed-to-modeled OH (left) and HO$_2$ (right) with the no-het model, as a function of altitude for the three regions: Colorado, Texas/Oklahoma, and Alabama. Dotted vertical black lines on right are approximate indicators of observation and model agreement.

[Figure]

[Figure]

**Figure S65**. Observations versus the het model. Scatter plots of observed versus modeled OH (upper right), HO₂ (upper left), HO₂/OH (lower left), and OH reactivity (lower right). Gray points are one-minute averages. Dashed red lines are factors of 1/1.4 and 1.4 times the fitted line (solid red).

[Figure]

[Figure]

**Figure S76.** Observations versus the hetmax model. Scatter plots of observed versus modeled OH (upper right), HO₂ (upper left), HO₂/OH (lower left), and OH reactivity (lower right). Gray points are one-minute averages. Dashed red lines are factors of 1/1.4 and 1.4 times the fitted line (solid red).

[Figure]

[Figure]

[Figure]

**Figure S87.** Median observed-to-modeled OH (top) and HO₂ (bottom) versus ice surface area concentration. Dotted horizontal black lines are indicators of observation and model agreement.

[Figure]

[Figure]

[Figure]

**Figure S98**. Calculated ozone production (P(O₃)). Rates of production (P) and loss (L) are on the left. Individual rates are shown, along with the total. Net P(O₃) is shown on the right.

[Figure]

[Figure]

**Figure S109**. HO$_2$NO$_2$ and temperature as a function of altitude. MCM331 was run the reaction rate coefficient for HO$_2$+NO$_2$+M from JPL18 (Burkholder et al., 2015) and from MCMv3.1.1 (Saunders et al., 2003).

[Figure]

[Figure]

**Figure S10.** Median modeled HO$_x$ loss (right) and production (left) as a function of altitude. MHP is  methyl hydroperoxide. The HO$_2$ heterogeneous loss was calculated with the het model.

**Table S1**. Scatter plot statistics for OH and HO$_2$ with ~10% of total one-minute data (661/6817)

| case | molecule | units | slope | intercept | R$^2$ | ratio |
|------|----------|-------|-------|-----------|-------|-------|
| no-het | OH | 10$^6$ cm$^{-3}$ | 0.93 | 0.20 | 0.78 | 1.08 |
| | HO$_2$ | pptv | 1.01 | 0.50 | 0.80 | 1.14 |
| het | OH | 10$^6$ cm$^{-3}$ | 1.05 | 0.21 | 0.84 | 1.1 |
| | HO$_2$ | pptv | 1.19 | 0.75 | 0.74 | 1.2 8 |
| hetmax | OH | 10$^6$ cm$^{-3}$ | 1.28 | 0.21 | 0.81 | 1.2 9 |

| | HO$_2$ | pptv | 1.68 | 0.83 | 0.59 | 1.87 |
|---|---|---|---|---|---|---|

---

## Editor Decision (ED1)

**Some comments/Requests from the Editor:**

**Referee 1.**

**In response to the referee you say:**

"The observed-to-modeled HO2 ratio shows little evidence of a NO-dependence. The reanalysis of Olson et al. (2006) explains the NO-dependence of the ratio discussed in Faloona et al., (1999), and the NO-dependence observed often in ground-based studies is not evident until NO exceeds 3-5 ppbv. Because the highest NO mixing ratio encountered in DC3 was about 3 ppbv, we would not expect to see this effect in the DC3 results."

**Editor:** Could you please give some very brief details in the revised MS about how the "re-analysis" by Olson et al. 2006 "explains" the NOx dependence of the measured/modelled HO2 ratio. Could you also please cite the other studies where substantial deviations observed during ground-based studies typically are not evident until NO exceeds 3 ppbv.

**For your response to another of the comments:**

"We agree that this is an interesting study to do, but it not within the scope of this paper. However, we did a quick check on this idea by calculating the observed-to-modeled HO2 ratio, filtering it for altitude bins (i.e., 5-8 km, 8-10 km, >10 km), and solar zenith angle bins (i.e., 50o-60o, 60o-70o, 70o-80o), and then plotting it as a function of J(O(1D)). In all cases, the ratio was constant as a function of J(O(1D)). This observation indicates that using the measured photolysis frequencies accurately captures the HOx production by photolysis as seen in Figure 4e."

**Editor:** Could you incorporate aspects of this response into the revised MS as it is an interesting finding you have stated.

**Referee 1 also says:**

page 8, line 6: The table showing the measurements is missing. It should be presented in the main paper. Accuracies and time resolution of the measurements should be given.

**Editor:** I agree with the referee that the Table belongs in the main paper. The table includes for example accuracy and time resolution of the main measurements which are modelled (OH, HO2) and the species included in the model (e.g. VOCs) – these are central pieces of information. Also in Table 1 it is not clear what the references mean, as some of these are quite old, e.g. 1991. Could you please indicate in the caption what the reference actually refers to? For example, is it a description of the instrument used on the aircraft for the measurements? (this is not clear owing to the date of the reference being quite old in in some cases)

**Referee 1 makes a point about there being additional VOCs in the convective outflows following rapid uplift…**

*…. is especially relevant in the outflow of convective clouds which can transport relatively short-lived species to higher altitudes in short time.*

**Referee 2** *makes a similar comment about convective outflows where this may impact on the interference seen in the ATHOS instrument:*

*"On page 6, the OH and HO2 interferences suffered by ATHOS are discussed and the authors state that interferences are only significant above forests and cities and are negligible outside of the planetary boundary layer. The authors should comment on whether this statement remains valid when sampling in and around convective clouds which can rapidly transport air from the surface. The correction applied to the HO2 observations to account for RO2 interferences is rudimentary and more details should be provided on the type of RO2 species present. The correction will change with changing RO2 species present and this relies on model predictions for RO2 which is far from ideal"*

**Editor:** Although in your response you discuss the method you use for correcting for RO2 interferences, the MS still states that interferences are only significant just above forests and cities and is negligible above the PBL. Can you please modify this statement to say that interferences may also be important in the convective outflow at higher altitudes where VOC concentrations are higher owing to rapid uplift.

**Referee 2 says:**

*Pg 14, line 1: What was recalibrated in the instrument? The phototube? The authors should comment on why they think it is appropriate to apply a calibration performed so long after the campaign.*

**Your response:**

Looking back through the notes for DC3, we noticed that the procedure had not been followed completely and that the transmission of the window between the calibration lamp and the calibration flow tube, which is 0.86, had not been included in the calibration. We re-measured the window transmission and confirmed that it was still 0.86 and then applied this number to the OH and $HO_2$ calibration.

**Editor:** Please give some brief detail on the reason for recalibration in the revised MS as readers will be wondering the same as the reviewer.

**Editor:**

Other comments:

In the revised MS, you have the text:

We have rewritten Section 4.2, 2nd paragraph as
"In DC3, the DC-8 spent hours flying in anvils of the cumulus clouds, which consisted of ice particles. DC3 provides evidence that the $HO_2$ uptake on ice is small. These results are consistent with $HO_2$

results over the western Pacific Ocean (Olson et al., 2004) but not with those over the northern Atlantic (Jeaglé et al., 2000). In Mauldin et al. (1998), a large difference between the observed and modeled OH was found in clouds, but this difference may have been due to the lack of photolysis frequency measurements, which are crucial to test photochemistry in a cloudy environment. In DC3, the DC-8 spent essentially no time in liquid clouds, for which there is evidence of measurable $HO_2$ uptake (Olson et al., 2006; Whalley et al., 2015). Thus these DC3 results provide constraints of $HO_2$ uptake on aerosol and ice particles, but not on liquid water particles."

**Editor:** The following paper ought to be cited and briefly discussed here, as it shows that the HO2 concentration is reduced when the aircraft sampled clouds, and the concentration was reduced more than expected owing to the reduction in J(O1D), providing some evidence for heterogeneous uptake which was not included in the calculation of [HO2]. The liquid water measurements shows these are primarily liquid clouds.

Commane, R., Floquet, C. F. A., Ingham, T., Stone, D., Evans, M. J., and Heard, D. E.: Observations of OH and $HO_2$ radicals over West Africa, Atmos. Chem. Phys., 10, 8783-8801, https://doi.org/10.5194/acp-10-8783-2010, 2010.

**Also:**

""Modeled $RO_2$ is primarily $CH_3O_2$ and $CH_3CHO_2$ above 5 km."

Change to $CH_3CH_2O_2$

---

## Author Response (AR2)

Response to reviewer's comments – second iteration

*Some comments/Requests from the Editor:*

*Referee 1.*

*In response to the referee you say:*

*"The observed-to-modeled HO2 ratio shows little evidence of a NO-dependence. The reanalysis of Olson et al. (2006) explains the NO-dependence of the ratio discussed in Faloona et al., (1999), and the NO-dependence observed often in ground-based studies is not evident until NO exceeds 3-5 ppbv. Because the highest NO mixing ratio encountered in DC3 was about 3 ppbv, we would not expect to see this effect in the DC3 results."*

*Editor: Could you please give some very brief details in the revised MS about how the "re-analysis" by Olson et al. 2006 "explains" the NOx dependence of the measured/modelled HO2 ratio. Could you also please cite the other studies where substantial deviations observed during ground-based studies typically are not evident until NO exceeds 3 ppbv.*

We expanded this discussion to include results from INTEX-A and moved the discussion from section 3.1, last paragraph to section 4.1 Comparing DC3 to Previous Studies, second-to-last paragraph.

"For DC3, observed and modelled $HO_2$ appear to agree as a function of NO up to about 2 ppbv, which are the highest NO values encountered. For several previous ground-based studies, the observed $HO_2$ was not obviously greater than the modelled $HO_2$ until NO reached ~2 ppbv or greater (Martinez et al., 2003; Ren et al., 2003; Shirley et al., 2006; Kanaya et al., 2007; Brune et al., 2016). For aircraft studies, in some cases the observed $HO_2$ did not obviously exceed the modelled $HO_2$ until NO approached 2 ppbv (Bair et al, 2017), while in other studies, the obvious exceedance occurred when NO was only a few hundred pptv (Faloona et al., 1999; Ren et al., 2008). Olson et al. (2006) showed that the Faloona et al. (1999) results for the SUCCESS campaign could be explained by the averaging of sharp plumes containing high NO and depleted $HO_2$ with the surrounding air. They showed that the SONEX results could be mostly explained by including all $HO_x$ precursor observations and updated kinetic rate coefficients and photolysis frequencies in the model. For INTEX-A (Ren et al., 2008), the enhanced NO is in the upper troposphere, where the observed-to-modeled $HO_2$ reached a factor of 3. It is possible that the $HO_2$ calibration was in error at low pressure (i.e., higher altitudes), although observed and modelled $HO_2$ agree in the stratosphere. It is also possible that there were missing $HO_2$ sources or outdated reaction rates in the model chemistry. We intend to re-examine INTEX-A and other previous NASA DC-8 missions that included ATHOS to see if an updated model can better simulate these $HO_2$ observations. "

*For your response to another of the comments:*

*"We agree that this is an interesting study to do, but it not within the scope of this paper. However,*

*we did a quick check on this idea by calculating the observed-to-modeled HO2 ratio, filtering it for*

*altitude bins (i.e., 5-8 km, 8-10 km, >10 km), and solar zenith angle bins (i.e., 50o-60o, 60o-70o, 70o80o),*

*and then plotting it as a function of J(O(1D)). In all cases, the ratio was constant as a function*

*of J(O(1D)). This observation indicates that using the measured photolysis frequencies accurately*

*captures the HOx production by photolysis as seen in Figure 4e."*

*Editor: Could you incorporate aspects of this response into the revised MS as it is an interesting*

*finding you have stated.*

In the last paragraph in section 3.1, we have rewritten it to include this observation and an observations of differences in observed and modeled $HO_2$ that was found for $O_3$.

"Another good test of the model photochemistry is the comparison of measured and modeled OH and $HO_2$ as a function of controlling variables (Fig. 4). The photolysis frequency for $O_3$ producing an excited state O atom, $JO(^1D)$, and $O_3$ are both involved in the production of OH. $O_3$ and NO cycle $HO_2$ to OH, while modeled OH reactivity cycles OH back to $HO_2$. In general, measured and model OH and $HO_2$ agree from $2x10^{-6}$ $s^{-1}$ to $7x10^{-5}$ $s^{-1}$ for $JO(^1D)$, from $2x10^{-3}$ ppbv to $7x10^{-1}$ ppbv for NO, from 40 ppbv to 100 ppbv for $O_3$. For $JO(^1D)$ greater than $2x10^{-5}$ $s^{-1}$, the median observed-to-modeled $HO_2$ ratio is 0.98; the in-cloud ratio is an insignificant 10% less than in clear air, indicating that the observed photolysis frequency measurement is accurate even in clouds. The observed-to-modeled $HO_2$ ratio shows little evidence of a NO-dependence, although observed-to-modeled $HO_2$ exceeded 2 for ~2% of the values when NO was more than 0.5 ppbv. For the $O_3$ observations greater than 200 ppbv, which are 0.5% of all observations, the observed-to-modeled $HO_2$ and OH are both ~0.5. It is possible that the behavior as a function of controlling variables is also a function of altitude. However, with the exception of low values of $JO(^1D)$, the median observed-to-modeled OH and observed-to-modeled $HO_2$ are generally independent of both the controlling variables and altitude (Fig. S4). The observed-to-modeled OH and $HO_2$ are also independent of whether the measurements were made in Colorado, Texas/Oklahoma, or Alabama (Fig. S5), although the ratios for some altitudes vary widely due to fewer data points in the altitude medians."

*Referee 1 also says:*

*page 8, line 6: The table showing the measurements is missing. It should be presented in the main*

*paper. Accuracies and time resolution of the measurements should be given.*

*Editor: I agree with the referee that the Table belongs in the main paper. The table includes for*

*example accuracy and time resolution of the main measurements which are modelled (OH, HO2)*

*and the species included in the model (e.g. VOCs) – these are central pieces of information. Also in*

*Table 1 it is not clear what the references mean, as some of these are quite old, e.g. 1991. Could you*

*please indicate in the caption what the reference actually refers to? For example, is it a description*

*of the instrument used on the aircraft for the measurements? (this is not clear owing to the date of*

*the reference being quite old in in some cases)*

We feel that this table belongs in the Supplementary Material, but will move it.

We thought that it was clear that the Reference was the reference describing the instrument, but have added the words "that describes the instrument" to the heading. Some references are quite old because the technique and the core instruments are quite old but have been constantly improved.

*Referee 1 makes a point about there being additional VOCs in the convective outflows following*

*rapid uplift…*

*…. is especially relevant in the outflow of convective clouds which can transport relatively short-lived*

*species to higher altitudes in short time.*

*Referee 2 makes a similar comment about convective outflows where this may impact on the*

*interference seen in the ATHOS instrument:*

*"On page 6, the OH and HO2 interferences suffered by ATHOS are discussed and the authors state*

*that interferences are only significant above forests and cities and are negligible outside of the*

*planetary boundary layer. The authors should comment on whether this statement remains valid*

*when sampling in and around convective clouds which can rapidly transport air from the surface. The*

*correction applied to the HO2 observations to account for RO2 interferences is rudimentary and more*

*details should be provided on the type of RO2 species present. The correction will change with*

*changing RO2 species present and this relies on model predictions for RO2 which is far from ideal"*

*Editor: Although in your response you discuss the method you use for correcting for RO2*

*interferences, the MS still states that interferences are only significant just above forests and cities*

*and is negligible above the PBL. Can you please modify this statement to say that interferences may*

*also be important in the convective outflow at higher altitudes where VOC concentrations are higher*

*owing to rapid uplift.*

In section 2.2, 4th paragraph, we add a sentence capturing these concerns.

"On the other hand, the deep convective clouds encountered in DC3 can lift short-lived VOCs that cause the $HO_2$ interference to the upper troposphere. Because the ATHOS was still sensitive to this $RO_2$ interference in DC3, we are not able to determine if this interference is affecting the $HO_2$ observations around and in these clouds."

*Referee 2 says:*

*Pg 14, line 1: What was recalibrated in the instrument? The phototube? The authors*

*should comment on why they think it is appropriate to apply a calibration performed so long after the campaign.*

*Your response:*

*Looking back through the notes for DC3, we noticed that the procedure had not been followed completely and that the transmission of the window between the calibration lamp and the calibration flow tube, which is 0.86, had not been included in the calibration. We re-measured the window transmission and confirmed that it was still 0.86 and then applied this number to the OH and HO2 calibration.*

*Editor: Please give some brief detail on the reason for recalibration in the revised MS as readers will be wondering the same as the reviewer.*

In section 3.5, 3rd paragraph, we add a sentence giving this reason.

"This recalibration was needed to account for the window absorption of calibrated 185 nm radiation that was neglected in the initial DC3 calibration."

*Editor:*

*Other comments:*

*In the revised MS, you have the text:*

*We have rewritten Section 4.2, 2nd paragraph as*

*"In DC3, the DC-8 spent hours flying in anvils of the cumulus clouds, which consisted of ice particles. DC3 provides evidence that the HO2 uptake on ice is small. These results are consistent with HO2 results over the western Pacific Ocean (Olson et al., 2004) but not with those over the northern Atlantic (Jeaglé et al., 2000). In Mauldin et al. (1998), a large difference between the observed and modeled OH was found in clouds, but this difference may have been due to the lack of photolysis frequency measurements, which are crucial to test photochemistry in a cloudy environment. In DC3, the DC-8 spent essentially no time in liquid clouds, for which there is evidence of measurable HO2 uptake (Olson et al., 2006; Whalley et al., 2015). Thus these DC3 results provide constraints of HO2 uptake on aerosol and ice particles, but not on liquid water particles."*

*Editor: The following paper ought to be cited and briefly discussed here, as it shows that the HO2 concentration is reduced when the aircraft sampled clouds, and the concentration was reduced*

*more than expected owing to the reduction in J(O1D), providing some evidence for heterogeneous uptake which was not included in the calculation of [HO2]. The liquid water measurements shows these are primarily liquid clouds.*

*Commane, R., Floquet, C. F. A., Ingham, T., Stone, D., Evans, M. J., and Heard, D. E.: Observations of OH and HO2 radicals over West Africa, Atmos. Chem. Phys., 10, 8783-8801, https://doi.org/10.5194/acp-10-8783-2010, 2010.*

We added the reference. In the Introduction we add the following sentence to paragraph 5.
"During the African Monsoon Multidisciplinary Analyses (AMMA) campaign, daytime $HO_2$ observations were generally simulated with a photochemical steady-state model, but not in clouds, where modeled $HO_2$ greatly exceeded observed $HO_2$ (Commane et al.,2010), suggesting $HO_2$ uptake on liquid cloud drops."

*Also:*
*""Modeled RO2 is primarily CH3O2 and CH3CHO2 above 5 km."*
*Change to CH3CH2O2*
Typo fixed.